# Competitive advantage, relationship, and benefit: Primary and secondary influencing factors of supply chain collaboration in China's advanced manufacturing industry

**Haohua Liu[1], Lin Lin[1]\*, Yang Cheng[2], Xiuling Chen[1,3], Jianwei Ren[4]**

**1** School of Business Administration, Jiangxi University of Finance and Economics, Nanchang, Jiangxi, China, **2** Department of Materials and Production, Aalborg University, Aalborg, Denmark, **3** Business School, Wuyi University, Nanping, Fujian, China, **4** Transportation Institute, Inner Mongolia University, Hohhot, Inner Mongolia, China

\* 997712454@qq.com

## Abstract

The advanced manufacturing industry is located at the top of the manufacturing value chain. Its development is restricted by supply chain collaboration (SCC), the level of which is affected by many factors. Few studies comprehensively summarize what influences SCC and distinguish the impact level of each factor. Practitioners have difficulty isolating the primary factors that affect SCC and managing them effectively. Therefore, based on synergetics and the theory of comparative advantage, this study analyzes what influences SCC in the advanced manufacturing industry and how these influencing factors work, using data from 94 manufacturing enterprises and the Haken model to identify the influencing factors. The results show that China's advanced manufacturing supply chain underwent a phase change and entered a new stage during 2017–2018. In the new stage, the competitive advantages of enterprises are one order parameter (slow variable) and are primary factors affecting SCC. The interest demands of enterprises are a fast variable and are secondary factors affecting SCC. The competitive advantages of enterprises dominate the interests of enterprises in affecting the collaboration level of China's advanced manufacturing supply chain. In addition, in the process of influencing SCC, there is a positive correlation between the competitive advantages of enterprises and the interest demands of enterprises, and the two factors have a positive feedback mechanism. Finally, when the enterprises in the supply chain cooperate based on their differential advantages, the collaboration capability of the supply chain is at the highest level, and the overall operation of the supply chain is orderly. In terms of theoretical contribution, this study is the first to propose a collaborative motivation framework that conforms to the characteristics of sequential parameters, which provides a theoretical reference for subsequent studies on SCC. In addition, the theory of comparative advantage and synergetics are linked for the first time in this study, and both of them are enriched and developed. Equally importantly, this study compares the bidirectional influence between firms' competitive advantages and firms' interest demands and the ability of both to influence SCC, enriching previous validation studies of unidirectional influence. In

**Data Availability Statement:** All relevant data are within the paper and its Supporting Information files.

**Funding:** Haohua Liu was funded by the Jiangxi Provincial Social Science "14th Five-Year Plan" (2021) Fund Project (grant no. 21YJ22D) and the Jiangxi University Humanities and Social Science Project (grant no. GL20112). Xiuling Chen was funded by the Social Science Planning Project of Fujian Province (grant no. FJ2021X017), and Jianwei Ren was funded by the National Natural Science Foundation of China (grant no. 71862026). Haohua Liu and Xiuling Chen contributed to the study design and content description. Jianwei Ren, who was one of the funders, contributed to the revision of the manuscript.

**Competing interests:** The authors have declared that no competing interests exist.

terms of practical implications, this study guides top managers to focus on the management practice of collaborative innovation in the supply chain and advises purchasing managers and sales managers on selecting supply chain partnerships.

## 1. Introduction

In facing the challenges of the new industrial revolution era, the world's manufacturing power-houses and developed countries have adjusted their national strategies and policies to promote advanced manufacturing and science and technology [1]. The COVID-19 pandemic has made manufacturing a costly industry [2]. Industries are interconnected in the manufacturing supply network, and a disruption in one of the supply chains can affect the operations of multiple industries [3]. Companies in the same industry may also work together as suppliers/distributors in a supply chain. There may be situations where these suppliers are supplying goods back and forth to each other. Hence, if one supplier has operational problems, the number of affected supply chain participants is bound to be large in such a situation where supply chain structures overlap. If a company is only in one supply chain, the supply chain participants also have to face huge operational pressure when the supply chain is hit. As a segment of the manufacturing industry, the development of the global advanced manufacturing industry is plagued by supply chain problems at present. Although China is making increased efforts to develop its advanced manufacturing industry, it is still lagging behind the world level in some fields, especially in the field of high-end industrial software, semiconductor processing equipment, materials, chips, and other high-end core technologies, which are in the situation of being "necked" by developed countries [4]. In addition, the proportion of advanced manufacturing in China's manufacturing industry is still small, and the economic benefits and profits obtained by it are not obvious. Faced with these internal and external problems, China's advanced manufacturing industry must solve the challenges of insufficient competitive advantage, obstructed product supply, and low economic effect related to supply chain problems. Supply chain collaboration (SCC) has become an important option to consider.

SCC is deeply affected by the situation inside and outside the supply chain. Logistics disruptions and lack of capacity supply during the pandemic caused a series of operational problems. To optimize business processes, improve rapid response capabilities, increase profits reduced by stagnant sales, achieve close collaborative relationships, improve trust levels, and so forth, in 2021, Hisense and JD upgraded the traditional "production by order" to "production by plan" based on market demand through deep SCC such as planning collaboration, procurement, distribution and return data collaboration, and inventory layout optimization. This has greatly improved supply chain efficiency and met their goal of laying out SCC. As we can see, their numerous purposes of laying out SCC influence their layout measures, which, in turn, influence the level of SCC they are able to achieve. Their ability to achieve a higher level of SCC is due to the fact that their layout measures are related to the main influencing factors of SCC. To effectively cope with the problems caused by their own shortcomings, trade frictions, and the COVID-19 pandemic, Chinese advanced manufacturing supply chains need to grasp the key to realizing SCC. In fact, it is still difficult to identify the main influencing factors of SCC. Synergetics does not specify the main factors affecting the synergy, but only point out that the sequential parameters are significant and rooted signs before and after the phase transition of the system and guide the fast variables to influence the synergy of the system [5]. Although, in recent years, sequential parameters have become the focus of many researchers to

analyze SCC [5, 6], many advanced manufacturing companies are helpless in the face of the unprecedentedly complex situation without theories showing the specific factors covered by the sequential parameters. Therefore, identifying the specific main factors affecting synergy becomes an important aspect of enriching the research related to sequential parameters in synergetics. In addition, previous research results have confirmed that factors affecting SCC, such as the level of information technology [7], environmental protection tendency [8], profit distribution mechanism [9], and cooperative strategies involving accountability and reporting mechanisms [10], are spread throughout the supply chain system. However, these redundant and scattered elements make it difficult to grasp the key direction of the development of SCC. Moreover, some researchers have determined the key influencing factors of SCC in manufacturing industry based on the characteristics of specific research subjects [6], but this approach is not universal. Therefore, the construction of a theoretically instructive and widely applicable system of influencing factors in the field of collaboration has become the research focus of SCC. Furthermore, facing SCC in manufacturing industry, researchers mostly explore the existance of the influence of profit revenue, competitive advantages, and non-contractural relationships [11–14]. However, they neither pay attention to the comparison of these influencing capabilities, nor take into account the interaction between profits and competitive advantages, which are believed to have an impact on SCC. Therefore, the influence capabilities of synergistic influencing factors is not sufficiently explored. The purpose of this study is to identify the primary and secondary influencing factors of SCC and their optimal influencing states in China's advanced manufacturing industry. Therefore, the purpose of this study corresponds to the following questions:

1. What are the primary and secondary influencing factors of SCC in China's advanced manufacturing industry, and what are their relationship mechanisms?

2. In terms of their effect, what state of the main influencing factors will lead to the highest level of SCC in China's advanced manufacturing industry?

Current research in the field of synergetics has not yet pointed out the system of motives with wide applicability. Researchers mostly agree that profit revenue, competitive advantage, and non-contractual forms of relationship (e.g., trust) influence manufacturing SCC. However, they only note the existence of these influences without paying attention to the contrasts between them or considering the interaction of profit and competitive advantages when these two factors influence manufacturing SCC. Compared with the current literature, we have achieved a innovation in viewpoint, i.e., we have developed a triadic motivation framework of "competitive advantages—interest demands—cooperative relationships" based on the theory of comparative advantage and synergetics. We also have a perspective innovation, which is reflected in the fact that, unlike the previous validation studies on the existence of various factors influencing manufacturing SCC, we further compare the influence ability of various possible factors and find the priority of the influence ability. In detail, this study compares the influence of competitive advantages and interest demands on the synergy of advanced manufacturing supply chains and also compares the interaction forces of competitive advantages and interest demands. Therefore, this study has theoretical implications in three ways. The ternary motivational framework developed in this study is a pioneering approach to the role of sequential parameters in a given system and shows the characteristics of the influence of sequential covariates on the system in terms of influencing factors for the first time. In addition, the ternary motivational framework of this study also enriches and develops the theory of comparative advantage and synergetics. Finally, this comparative analysis made in terms of perspective innovation enriches both the previous one-way influence research between

interests and advantages and the single-channel analysis of the influence of interests and advantages on SCC. Most importantly, this comparative analysis highlights the main tone and overriding role of competitive advantage that influences SCC in advanced manufacturing and explains economic phenomena that have not been previously analyzed in the literature regarding the main influencing factors. The research content has practical guidance for top management decision makers, purchasing managers, and sales managers of enterprises in the advanced manufacturing supply chain. The top-level decision makers of enterprises need to focus on innovative communication, learning, and cooperation in supply chain management activities and be committed to completing the layout of SCC with the cultivation process of competitive advantage. In terms of developing long-term partnerships, purchasing and sales managers need to select partners with specific strengths in the industry based on market and corporate strategies.

According to synergetics and the theory of comparative advantage, this paper analyzes the motives and the realization and mechanism of influencing factors of SCC in China's advanced manufacturing industry. It uses the Haken model to quantitatively discover the fundamental/main and secondary influencing factors of SCC in China's advanced manufacturing industry and their influencing mechanisms. At the same time, it determines the optimal state of the main influencing factors and provides suggestions for the development of China's advanced manufacturing supply chain. This paper consists of seven parts. The first presents the background, the problem, and the significance of the study. The second introduces the current status of previous research based on relevant literature, which lays the foundation for the research in this study. The third part elaborates the evolutionary dynamics of SCC in the advanced manufacturing industry and the formation of influencing factors from a theoretical perspective. Then, hypotheses are formulated. The fourth introduces the model used in this study, constructs an evaluation index system for motivations of SCC in advanced manufacturing, and pre-processes the collected data. The fifth part is an objective interpretation based on the empirical results. The sixth discusses the empirical findings. The seventh draws the conclusions of this study and makes some suggestions for further research based on the research deficiencies. Through the above research framework, it is found that the primary and secondary influencing factors of SCC in China's advanced manufacturing industry are the competitive advantages and interest demands of enterprises respectively, which further promote each other. It is also shown that differential advantages among companies can achieve optimal SCC.

## 2. Literature review

This study explores the primary and secondary influencing factors of SCC in the advanced manufacturing industry based on the motivational framework developed by the theory of comparative advantage and synergetics and discovers the relationship mechanisms and effects of the primary and secondary influencing factors. Therefore, the relevant areas of this study are SCC influencing factors, relationships among influencing factors of manufacturing SCC, and research on the application of the theory of comparative advantage and synergetics.

### 2.1 Influencing factors of SCC

The structure of the supply chain is such that the influencing factors of SCC are complex and many of them have certain commonalities and can be grouped together.

Many researchers acknowledge that, since all stakeholders in a dynamically connected supply chain network are interconnected and dependent on each other [15], coordination of profit guarantees can be effective in maintaining a stable win-win situation among supply chain members in the long run and SCC should meet the profit objectives of supply chain members.

Huang et al. argued that profits of supply chain members are an important concern in the setting of SCC mechanisms [9]. Choi et al. proposed a way to optimize the supply chain cooperation mechanism by selecting common contracts to achieve as much as possible the cooperation expectations of retailers with desired profit targets [16]. In addition, much of the literature has looked at profit-generating means, such as increasing product value and meeting order demand, as the original purpose of SCC. Chauhan et al. suggested that, to achieve the common goal of maximizing the value of goods/materials, firms should exchange knowledge and collaborate in a mutually beneficial manner, thereby achieving the flow and utilization of resources between organizations [11]. Nayeri et al. found that, during the COVID-19 pandemic, to meet the urgent order demand, firms in the medical device manufacturing supply chain made full use of the synergy concept for supply chain governance [17]. The above studies reveal a series of profit-oriented motives for SCC. Profit-based behavior stems from interest claims [18, 19]. Interest demands of enterprises refer to an organization's desire for wealth and industry status to obtain survival and growth [20, 21]. However, some studies have not elevated profit-based behavior to the level of benefit claims when analyzing the impact on SCC. We took this into account in the design of this study.

Non-contractual forms of cooperative relationships such as trust, knowledge sharing, cooperative culture, and relational commitment have become the focus of many scholars studying SCC and are considered to be the key determinants of SCC [22]. Hall et al. found that supply chain responsiveness had been improved under the influence of collaborative culture [12]. As an example of collaborative culture, the provision of life-saving medical equipment and personal protective equipment in the event of an outbreak can help companies effectively improve the operational problems caused by COVID-19. In terms of trust, Berardi et al. argued that it is difficult to develop trusting relationships between firms but that the existence of trusting relationships contributes to the success and continuity of SCC [13]. Digitization enhances data and information exchange between partners, whose trace capability improves overall supply chain visibility and security. Therefore, through increased trust, digitization can improve collaborative relationships and maintain long-term stability of the partnership, and, in turn, improve the level of SCC [23]. Thus, emerging technologies such as big data are being widely used in collaborative supply chain management [24, 25]. In assessing the impact of COVID-19 on the supply chain in the healthcare industry, Bhaskar et al. proposed a coordinated governance system for the supply chain based on new technologies such as blockchain to enable intervention in emergencies [26]. Bumblauskas et al. found that blockchain technology can effectively facilitate the establishment of trust relationships between producers and consumers and that the business–consumer synergy that results from such trust relationships can facilitate the development of new products based on information such as user profiles [27]. Viriyasitavat et al. argued that digital technology–based supply chains are more intelligent and can achieve higher SCC such as ensuring product quality and reducing supplier risk [28]. Isolating the essence of the above findings reveals that non-contractual relationships such as collaborative culture, trust, and knowledge sharing between companies are, in fact, the concrete manifestation of the closeness of their cooperative relationships. Cooperative relationships between enterprises involve long-term cooperative agreements based on reciprocal commitments, and these agreements are often accompanied by resource transactions between subjects [29]. Non-contractual means of knowledge exchange, trust, and commitment among firms are rooted in the social relationship structures of firms, which effectively provide opportunities for firms to integrate internal and external knowledge and resources, generate trust, and establish commitment [30]. Therefore, the ties among supply chain members affect the dissemination of knowledge and management, the creation of trust, and the generation of commitment, and the implementation of corporate strategies and supply chain management of member firms may

be affected as a result [31]. Most of the literature on SCC only considers the impact of certain specific forms of partnerships. However, when considering the influences on SCC, the manifestations of cooperative relationships are not actually limited to trust, knowledge exchange, and cooperative culture. To ensure a wide range of study coverage, this paper therefore digs deeper and explores the impact of corporate partnerships on SCC.

Long et al. argued that green manufacturing closed-loop SCC is inevitably influenced by the green attitudes and behaviors of multiple stakeholders [8]. Yang et al. argued that logistics bottlenecks in logistics-e-commerce complex systems affect collaboration between logistics and e-commerce [6]. Silbernagel et al. suggested that, to reduce global related costs and significantly improve product quality, many companies are keen on SCC [14]. Viriyasitavat et al. argued that manufacturing firms can selectively integrate services into IoT-based business process management to achieve distributed dynamic collaboration management, thereby enhancing the efficiency of SCC [32, 33]; in addition, to apply excellent production management practices and expand the company's capabilities, such as flexibility in manpower and material mobilization [34], firms often tend to improve the level of SCC [35]. The degree of low carbon and environmental friendliness of products, logistics efficiency, product quality, and production efficiency mentioned in the above literature are concrete manifestations of competitive advantage. Competitive advantages of enterprises are the favorable conditions that a firm has over other firms in terms of organizational structure, production scale, product quality, product development, productivity, branding, and reputation [32]. Luthra et al. argued that the formation of competitive advantage is influenced by the resources and capabilities required by the organization [34]. Thus, in the face of global trade and the use of digital technologies, closure can put firms and their supply chain partners at a competitive disadvantage due to the relative limitation of the resources available to them [18]. Facing this, firms engage in SCC to achieve the integration of rare or irreplaceable resources/capabilities and gain additional competitive advantages [34]. The above studies show that many scholars have only thought about the influencing factors of SCC in a one-sided way. The current literature is unable to present a comprehensive and complete list of influencing factors.

## 2.2 Relationship between factors influencing SCC in manufacturing

Based on the results of the literature review above, this study focuses on the following influencing factors: competitive advantage, interest claim, and partnership.

The relationship between the interests, cooperation, and advantages of firms in manufacturing supply chains has received much attention from researchers. Obviously, collaborative relationships among supply chain partners are considered to be the basis for the development of related activities aimed at shaping competitive advantage. Tao et al. argued that due to the complexity and high risk of innovation, innovation activities are gradually moving toward multi-domain and multi-subject combinations, and, therefore, SCC can facilitate technological innovation [36, 37]. For example, Nike not only helped its suppliers to improve the overall quality and efficiency of their operations, but also reduced the negative impact of the poor working conditions in its suppliers' factories by empowering workers [38]. There are also studies that show the impact of collaborative relationships on business benefits. SCC is one of the core ways to improve the business performance of firms. For example, trusting relationships among supply chain members help to reduce the level of detail required in contracts, reduce the need for monitoring, and, consequently, reduce the associated costs [39]. In addition, as Silbernagel et al. pointed out, through SCC, firms can exchange data and information within a global production network, and the increased transparency from this can reduce unnecessary costs [33]. For example, in the case of advertising cooperatives formed by

suppliers, manufacturers, and retailers, the market share of the supply chain is expanded and the overall supply chain revenue is increased [40]. Benefits can have a direct or indirect impact on the cooperative relationships of firms. In terms of direct effects, Jiang et al. argued that benefit distribution mechanisms affect the stability of technological innovation cooperative alliances formed by supply chain firms [41]. Benefits of business partnerships can be indirectly contributed through firm competitiveness. Thanh et al. found that firm competitiveness is an important dimension that promotes improvement in all aspects of firm performance [42, 43]. According to Ghadge, in the face of the pressures brought about by the global economy, industries need to seek supply chain partners with the goal of maximizing benefits and catering to high customer expectations, which requires firms to use the concept of association to strive for profit maximization, thus maintaining dynamic competitiveness [40]. Taken together, these studies reveal that the interests, competitive advantages, and competitive relationships of firms in the supply chain are closely linked. However, there is no literature that provides a clear answer to the ranking of the magnitude of influence between these factors in the supply chain.

## 2.3 Application of the theory of comparative advantage and synergetics

The process of achieving energy and matter exchange between a system and its external environment is subject to the risk of imbalance [44]. The German scholar Haken's concept of synergetics focuses on how complex open systems spontaneously form ordered structures through coordination or antagonistic effects within the system in the face of imbalance risks. Using synergetics, Chinese scholars have solved many practical problems, such as understanding synergistic effects of industrial complex systems [5], developing specific strategies based on the synergistic effects of water allocation systems [45], and exploring the welfare effects and regional heterogeneity of ecological synergy in regional development [46]. Some researchers have also considered the sequential parameters, which play a major role in influencing order in synergetics, as synergistic influences on composite systems [6], but these researchers did not use the qualities of the sequential parameters as an assessment criterion when determining the influences on the synergy of system [6].

In economic systems, this orderliness is manifested in the synergy of the resources, elements, and benefits of the components [47]. Resources, elements, and benefits are also reflected in the theory of comparative advantage. The theory of comparative advantage essentially states that, in the face of international trade and the gap in productivity levels between countries, countries tend to export products with comparative advantage and import products with comparative disadvantage [48]. This allows each country to save labor, gain specialization, increase production efficiency, and ensure that each country maximizes the benefits of its own product market [49]. The theory of comparative advantage is now widely used in the field of international/regional economic trade [50]. For example, Shen et al. argued that the dynamics of comparative advantage in the structure of factor endowments in each region has an impact on global trade imbalances [51]. Dvoskin et al. argued that regional means of production influence the direction of trade through "chains of comparative advantage" [46]. Combined with the above analysis, the theory of comparative advantage and synergetics are linked in many ways. Synergetics is applicable in all systems, and the phenomenon of trade exchange between countries due to the difference in productivity levels mentioned in the theory of comparative advantage belongs to the socioeconomic system. In terms of application scenarios, both theories can be used in the study of socioeconomic systems. The theory of comparative advantage emphasizes the use of the advantageous gap in resource endowments between subjects, trade exchange relations (which ensure the flow of factors such as commodities), and the acquisition of greater market benefits for each subject to achieve hand-in-hand development [49]. In

terms of content points, this coincides with the essence of the synergy of economic systems (resources, factor flows, benefits) in the theory of synergy [47]. In other words, both theories analyze the collaboration, resources, factor flows, and benefits between multiple subjects. In terms of the role of content points, the nature of the synergy in socioeconomic systems determines the creation of synergistic phenomena. This role is also revealed in the theory of comparative advantage, where the occurrence of trade collaboration between countries is driven by the gap in resource endowments, the specialized division of labor and efficiency improvements generated by factor flows, and the maximization of market benefits [49]. Capturing the commonalities of the above three aspects, the theory of comparative advantage and synergetics can be combined. However, the current literature does not do so.

### 2.4 Research gaps

As mentioned above, scholars have explored the theoretical aspects of synergetics and influencing factors in SCC. These rich research results have provided good references for this paper. However, the literature also suffers from three shortcomings. First, in the literature of influencing factors, no scholars have mentioned interest claims, and many scholars only think at the level of profit. For example, profit is the purpose of SCC and profit win-win is the form of SCC [5, 11]. No scholars have proposed a comprehensive and systematic framework of influencing factors of SCC. Therefore, all the influencing factors of advanced manufacturing SCC cannot be determined. Second, some scholars have selected influencing factors according to the operational characteristics of specific research subjects [6, 8]. But this approach has the problem of limited scope of reference, and such apparent reasoning is contrary to the underpinning logic of "dominating the system to produce qualitative changes" of sequential parameters. The elements emphasized in the theory of comparative advantage are the same as those emphasized in the synergistic nature of the economic system in the study of synergetics [47, 48], but there is no research to point out and use the relationship between the two theories. Therefore, the current way of determining the influencing factors of advanced manufacturing SCC does not take into account the requirements of synergetics for the existence of their influencing capabilities. Third, although researchers have suggested that corporate interests and the manifestations of cooperative relationships interact [36, 39–41] and can influence manufacturing SCC separately [9, 16, 35], no literature has yet distinguished and analyzed the primary and secondary effects of interests, competitive advantage, and cooperative relationships on SCC. The mutual influence among competitive advantage, interest claims, and cooperative relationships is also not clearly ranked by the literature. Therefore, the key to upgrading the level of SCC in the advanced manufacturing industry cannot be determined. These shortcomings leave room for this study to develop a theoretically grounded influencing factor system of SCC that is consistent with the qualities of the covariates that trigger synergistic effects, complete and comprehensive for the first time, and with broad applicability. They also leave room to clarify for the first time the focal direction of SCC level upgrading in the advanced manufacturing industry through the differentiation ability of synergistic influencing factors.

## 3. Theoretical analysis

A review of the literature shows that the commonalities between the theory of comparative advantage and synergetics in terms of application scenarios, theoretical content, and the role of theoretical points reveal that the driving basis for effective collaboration among the components within a system is the gap in resource endowment advantage, factor mobility, and profit maximization [47, 48]. The gap in resource endowment advantage manifests itself as a competitive advantage [32]. The goal of profit maximization is rooted in the existence of interest

claims [18, 19]. Factor flows are built on cooperative relationships [31]. With the above theoretical derivation, this study develops the driving factors of system collaboration, that is, competitive advantage, interest claim, and cooperative relationship.

In addition, the content of synergetics can effectively show the mechanism of the role of various influencing factors and the mechanism of relationships between and among those factors. In fact, each element within the system exerts a differential influence on the system. Once the control parameters from outside the system push the system past the linear instability point, the differential influence of each element in the system is amplified. At this point, the system is able to distinguish between fast and slow variables that affect the system. The slow variables can enslave and control the fast variables. Their micro-variation can produce the "butterfly effect," that is, trigger dramatic changes in other variables. Slow variables dominate the evolution of the system to a new ordered structure and allow the system to produce phase changes according to this principle. Therefore, slow variables are called the sequential parameters of the system [5] and can be considered the primary influences of synergy. In contrast, the fast variables are the secondary influences. When the aggregation state represented by the slow variable reaches a threshold, the whole system produces a synergistic effect. As a synergistic effect, the phenomenon of self-organization reveals that the system spontaneously brings the accumulated quantitative changes to qualitative effect under the synergistic action of the subsystems. At this point, there is a system phase transition; that is, when a substance reaches a threshold state, the phase changes from chaotic to stable and disorderly to orderly for a short period of time [52]. Therefore, the phase change is achieved when the system produces a synergistic effect.

To theoretically validate the rationality of the motivational framework, this study applies the theoretically developed motivational framework to the analysis of advanced manufacturing supply chains. In addition, the analysis of the generation and influence process of synergistic influencing factors is applied to the advanced manufacturing supply chain. This facilitates the execution and analysis of the empirical study.

## 3.1 Analysis of the motivation for the collaborative evolution of advanced manufacturing supply chains

The suppliers, manufacturing enterprises, processing enterprises, third-party logistics enterprises, and retailers in the supply chain of the advanced manufacturing industry cannot complete all the operations such as production, processing, inspection, transportation, and sales of products alone; therefore, the development of synergy becomes an inevitable choice for the supply chain of the advanced manufacturing industry. The fundamental function of the synergy of the system is to bring about the synergy of resources, elements, and benefits among its internal parts [47, 48]. First, the resources of enterprises in the supply chain are the basis for their participation in SCC and reflect their competitive advantages. The abundance of resources affects the quality of the development of the advanced manufacturing industry and is a key element to support the development of the advanced manufacturing industry. Advanced manufacturing enterprises join the supply chain alliance with their resources. Through the integration of supply chain resources, they gain competitiveness with the synergy effect of "1+1>2." Second, the flow of elements within the supply chain is in line with the synergistic connotation of sharing, which is a manifestation of an enterprise cooperation relationship. Enterprises in the supply chain achieve positive interaction through logistics, capital, technology, talent, and other factors. In this process, enterprises reach cooperation and open up new channels to improve their competitiveness. Last, to satisfy the interests of supply chain members, the distribution of benefits requires a reasonable division of labor and collaboration

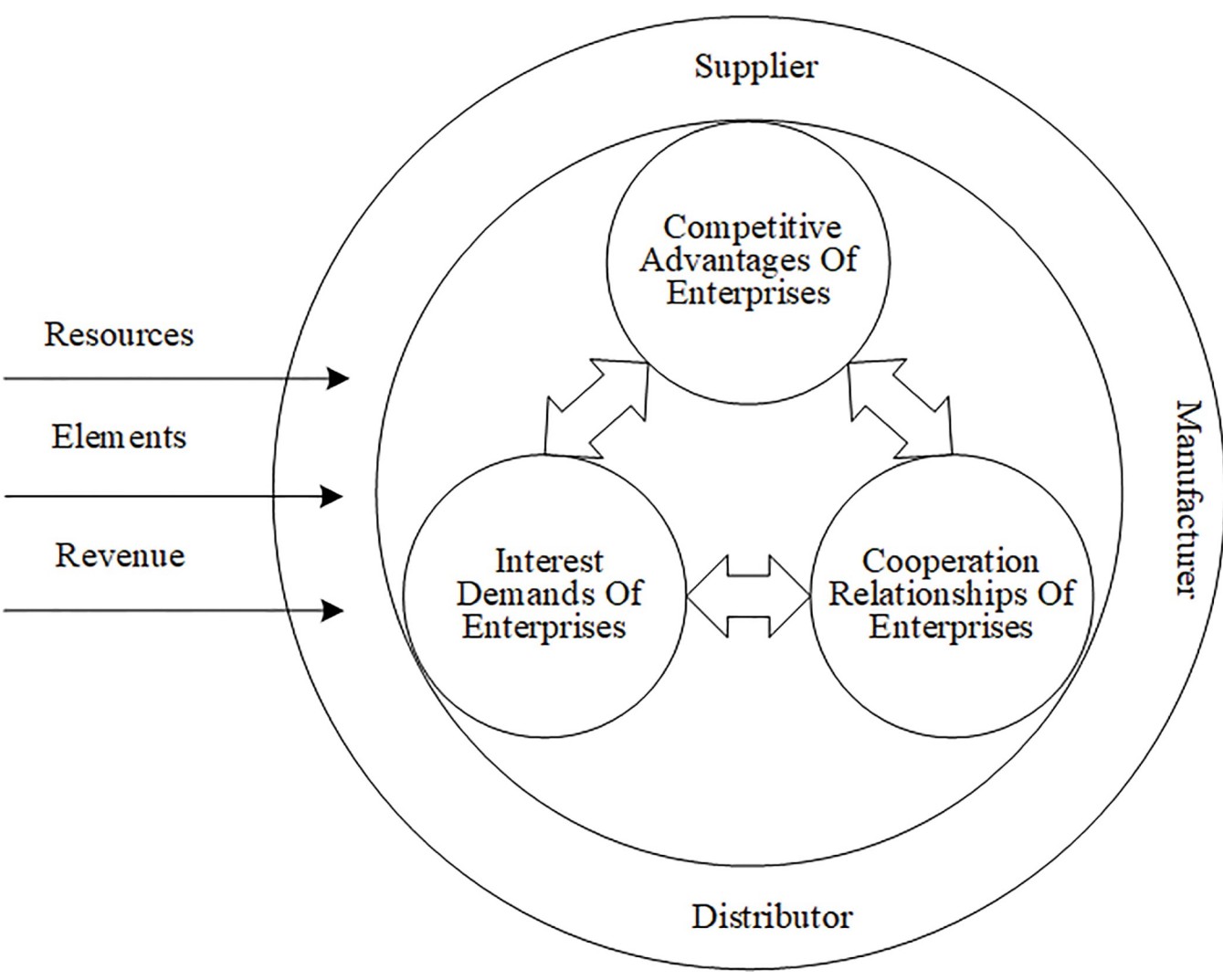

**Fig 1. Drivers of SCC evolution in advanced manufacturing.**

among companies in the supply chain. According to this division of labor and collaboration, supply chain members take on economic functions that are appropriate to their capabilities. Ultimately, the supply chain distributes benefits based on the contributions of its members. This cooperative benefit satisfies the expected benefits of the supply chain members and promotes the continuous cooperation of the supply chain members. The supply chain shapes competitive advantages with enterprise resources, strengthens the cooperative relationship between enterprises through the flow of elements within the supply chain, and allocates the corresponding share of benefits to enterprises in a division of labor and cooperation. As a result, the whole supply chain system achieves collaborative evolution. Based on the above analysis, the influencing factors of supply chain synergistic evolution can be divided into three aspects: competitive advantages of enterprises, cooperative relationships of enterprises, and interest demands of enterprises, as shown in Fig 1.

**3.1.1 Competitive advantages of enterprises.** Competitive advantages of enterprises in the supply chain are the internal driver of SCC and generate complementary effects through synergy. Cooperation based on competitive advantages will generate competitive spillover.

The Chinese advanced manufacturing industry is currently suffering from the problems of low technological innovation capability and lack of talent. There is an urgent need to improve innovation performance and competitiveness. The improvement of the level of innovation requires the help of inter-firm collaboration [53]. One reason for this is that the interaction and integration of resources between firms can improve their production procedures and achieve knowledge integration, thus promoting innovation in their processes and products [54]. The other reason is that the innovation performance of firms is also influenced by social performance. This is because innovations achieved by firms in collaborative processes can both improve social performance and obtain a greater level of innovation by this improved social performance [53]. Therefore, enterprises can focus their operation on competitive business and use SCC to integrate the complementary advantages of resources, capabilities, and technologies in the chain while playing their expertise. This will construct an additional competitive advantage that they do not have when operating alone.

**3.1.2 Cooperative relationships of enterprises.** Enterprise partnerships are the operational link of SCC. It is easier to establish long-term strategic partnerships between enterprises with a high degree of synergy. The cooperative relationship is formed by the flow of tangible and intangible elements. Tangible elements include products, talents, capital, and so forth, while intangible elements include information, knowledge, and so on. Facing a series of problems such as industrial structure to be optimized, poor development atmosphere, lack of industry standards, and insufficient international competitiveness, members of the Chinese advanced manufacturing supply chain need to strengthen communication and exchange and bring about strategic integrated collaboration based on product flow, capital flow, information flow, and talent flow. They need to transition the relationship between partners from one centered on production management to one centered on science and technology innovation. Through in-depth cooperation in new product development and design, information sharing, and market development, they can achieve collaborative innovation in supply chain technology and standards, improvement in resource elements, and an increase in market demand.

**3.1.3 Interest demands of enterprises.** The goal of SCC is to achieve the optimal profit of the supply chain and to ensure the sustainability of the interests of the enterprises in the chain. The business activities of each enterprise in the supply chain revolve around maximizing their interests. However, only the benefits based on cooperation are the most sustainable. The collaborative operation among the members of the supply chain can save the transaction cost in the process of cooperation, ensure the economy of rapid information communication, and reduce the total cost of supply chain operation. For advanced manufacturing, the supply process of enterprises often generates higher cost expenditures. Thus, the interconnection and collaboration of supply chain partners are critical to the profit growth of individual companies.

Competitive advantages of enterprises, cooperative relationships of enterprises, and interest demands of enterprises interact with each other in the process of supply chain collaborative evolution in the advanced manufacturing industry and jointly promote supply chain collaborative development [51, 53, 54]. First, interest demands of enterprises→cooperative relationships of enterprises→competitive advantages of enterprises. Under the influence of interest demands, advanced manufacturing enterprises establish cooperative relationships with their upstream and downstream enterprises to carry out production and operation activities. Through this, they integrate the resources between enterprises and enhance their competitiveness by integrating the competitive advantages of their partners. Second, competitive advantages of enterprises→interest demands of enterprises→cooperative relationships of enterprises. Competitiveness is the key to establishing the distribution of supply chain benefits. Improving the competitive ability of enterprises is conducive to improving the overall competitiveness of the supply chain, which can also ultimately optimize the distribution of supply

chain benefits. As a result, the enthusiasm of enterprises and their partners to participate in cooperation increases. Third, interest demands of enterprises→competitive advantages of enterprises→cooperative relationships of enterprises. The interest demand drives enterprises to look for partners with complementary competitive advantages. The competition of advantages between cooperating companies determines the stability of the supply chain cooperation relationship. Fourth, cooperative relationships of enterprises→interest demands of enterprises→competitive advantages of enterprises. In the cooperative alliance model, supply chain members achieve synchronized operation through information interaction. Hence, enterprises can reduce costs arising from inventory backlogs, improve the ability to deliver on time, and enhance the sensitivity of the production and sales to the market, forming a series of competitive advantages in the market.

### 3.2 Generation and operation mechanism of the influencing factors of phase change points in the collaborative evolution of the advanced manufacturing supply chain

Synergistic evolution is the form and means of self-organization. The phenomenon of self-organization reveals the spontaneous formation of a new ordered structure of the system according to a certain rule under the synergistic action of the subsystems. The phase change is a sudden change of the structure formed by the system that is compatible with the external environment under a particular condition (phase change point). The phase change is the precondition for the realization of the new ordered structure. As a management system, the advanced manufacturing supply chain, to realize phase change and form an orderly new structure through synergy, must carry out a self-organization process through the control of internal factors, as shown in Fig 2.

**3.2.1 Self-organization process 1.** The competitive results of synergistic motivations determine the influencing factors of phase change points in the collaborative evolution of the advanced manufacturing supply chain. Although the system evolution is jointly influenced by various factors external to the supply chain, the main evolutionary direction depends largely on the internal system. In other words, the system evolution is influenced by the competitive advantages, cooperative relationships, and interests of enterprises in the supply chain. The members of the advanced manufacturing supply chain establish a complex cooperative relationship based on common interests. Their ultimate goal is to quickly respond to customer demand, improve customer satisfaction, enhance the network externality effect of the supply chain, and realize the spontaneous gathering from mutual competition to synergy through self-organization without the intervention of external factors. To achieve this goal, the self-organization process of the advanced manufacturing supply chain system distinguishes three driving forces, namely, competitive advantages, cooperative relationships, and interest

**Fig 2. Self-organization process of advanced manufacturing supply chain.**

**Table 1. Hypotheses about the influencing factors of SCC in the advanced manufacturing industry.**

| Hypothesis | Content |
|---|---|
| H1 | For advanced manufacturing SCC, the competitive advantage of enterprises is the main influencing factor, and the interest demands of enterprises are the secondary influencing factor. |
| H2 | For advanced manufacturing SCC, the competitive advantage of enterprises is the main influencing factor, and cooperative relationships of enterprises are the secondary influencing factor. |
| H3 | For advanced manufacturing SCC, the interest demands of enterprises are the main influencing factor, and the competitive advantage of enterprises is the secondary influencing factor. |
| H4 | For advanced manufacturing SCC, the interest demands of enterprises are the main influencing factor, and cooperative relationships of enterprises are the secondary influencing factor. |
| H5 | For advanced manufacturing SCC, the cooperative relationships of enterprises are the main influencing factor, and the competitive advantage of enterprises is the secondary influencing factor. |
| H6 | For advanced manufacturing SCC, the cooperative relationships of enterprises are the main influencing factor, and interest demands of enterprises are the secondary influencing factor. |

demands, so that certain trends in the system can be strengthened and form the influencing factors that mark the orderly evolution of the system, that is, system parameters.

**3.2.2 Self-organization process 2.** The advanced manufacturing supply chain undergoes a phase change and orderly evolution. After the influencing factors of SCC are screened out in self-organization process 1, these internal influencing factors will cause phase changes in the supply chain system of the advanced manufacturing industry under certain circumstances and will influence and encourage the supply chain members to self-control, self-adjust, and interact with the environment. Eventually, the supply chain evolves in an orderly manner, that is, SCC is formed. At this time, the supply chain system turns into a collaborative system that operates according to the rules jointly formulated by all members and then realizes the supply chain upgrading.

Based on the above analysis, this study summarizes the possible scenarios for the results of the power differentiation of self-organization processes on competitive advantage, cooperative relationships, and interest claims. Accordingly, hypotheses are proposed and integrated into Table 1.

## 4. Methodology

### 4.1 Haken model

The selection of sequential parameters is the main problem of synergy [6, 55], and there are two approaches. Some scholars make subjective judgments based on the operational characteristics of a particular research subject; for example, Chen uses "traffic accessibility efficiency" as a sequential parameter in the "traffic-industry-city scale" system [56]. However, the use of this approach has domain-level limitations and is not research generalizable. In addition, according to synergetics, such phenotypic features may be the result of multiple variables acting together and are not fully representative of the variables acting. Another approach is to use the Haken model to find sequential parameters. The Haken model is derived from synergetics. Based on the principle of the dominance of sequential parameters over the system, the model mainly uses the adiabatic elimination method, that is, constructing the equations of motion and identifying the sequential parameters with the help of adiabatic approximation conditions. Therefore, the Haken model can identify the sequential parameters by the "differentiation of the influence of the factors at the point of linear instability" mentioned in synergetics. The Haken model has been widely used in various disciplines to explore the sequential parameters, such as the evolution analysis of industrial systems and supply chain systems, and has strong

applicability to this study [5]. Therefore, this paper draws on this model to identify the differentiated influence capabilities of SCC drivers in the advanced manufacturing industry and thus identify the sequential parameters and other influencing factors of SCC in the self-organization process.

The Haken model is actually a model in the field of physics [5]. It assumes that the parameter $q_1$ is the internal force of the whole system, and the internal force $q_1$ affects the orderliness of the whole system and controls the state of motion of the parameter $q_2$ [8]. Under the influence of $q_1$ and $q_2$, the equations of motion of the system can be expressed as:

$$\dot{q}_1 = -a_1 q_1 - b_1 q_1 q_2 \tag{1}$$

$$\dot{q}_2 = -a_2 q_2 + b_2 q_1^2 \tag{2}$$

The parameters in physics are continuous random variables. The application of physics Eqs (1) and (2) to the field of economic management requires the following discretization:

$$q_1(t+1) = (1-a_1)q_1(t) - b_1 q_1(t)q_2(t) \tag{3}$$

$$q_2(t+1) = (1-a_2)q_2(t) + b_2 q_1^2(t) \tag{4}$$

$a_1$ and $a_2$ denote the effects of $q_1$ and $q_2$ on the ordered state of the system, respectively. When $a_1$ is less than 1, it indicates a positive feedback mechanism with a gradual increase of $q_1$ within the system. When $a_2$ is less than 1, it indicates a positive feedback mechanism with a gradual increase of $q_2$ within the system. $b_1$ and $b_2$ reflect the synergistic effect of $q_2$ on $q_1$ and $q_1$ on $q_2$, respectively. When $b_1$ is less than 0, it indicates the facilitative effect of $q_2$ on $q_1$. When $b_1$ is more than 0, it indicates the inhibitory effect of $q_2$ on $q_1$. When $b_2$ is more than 0, it indicates the facilitative effect of $q_1$ on $q_2$. When $b_2$ is less than 0, it indicates the inhibitory effect of $q_1$ on $q_2$. The larger the absolute values of $b_1$ and $b_2$, the greater the effect.

The adiabatic approximation assumes that $|a_2| \gg |a_1|$ (usually requiring a difference of at least one order of magnitude) and $a_2 > 0$. This is based on the property that the sequential parameters are slow variables. Assuming that $q_1$ does not exist or disappears, $q_2$ will be reduced to 0 by the damping effect. This is because the damping of $q_1$ is much smaller than that of $q_2$. In other words, the rate of change of $q_2$ is much faster than the rate of change of $q_1$. Under the condition that the adiabatic approximation assumption is satisfied, if $q_2$ vanishes, $q_1$ is too late to change. Letting $\dot{q}_2 = 0$ in Eq (2), it follows that

$$q_2 = \frac{b_2}{a_2} q_1^2 \tag{5}$$

From Eq (5), it can be seen that $q_1$ governs, influences, and determines the change of $q_2$. Therefore, $q_1$ is the order parameter, which can influence the orderliness of the system. Combining Eqs (1) and (5), the evolution equation of the system under the domination of $q_1$ is obtained as follows:

$$\dot{q}_1 = -a_1 q_1 - \frac{b_1 b_2}{a_2} q_1^3 \tag{6}$$

The potential function describes the trajectory of the system and can determine the state of the system. The potential function is obtained by integrating the opposite of $\dot{q}_1$ in Eq (6):

$$V = \frac{1}{2}a_1 q_1^2 + \frac{b_1 b_2}{4a_2}q_1^4 \tag{7}$$

## 4.2 Sample selection, index system, and data processing

**4.2.1 Sample selection.** We take the advanced manufacturing supply chain as the research object. This study considers when phase changes occur in China's advanced manufacturing supply chain system and explores which factors affect SCC during the process and how they affect it. The Haken model requires precise measurement results. Questionnaire and structured interview methods are prone to missing and wrong answers and may result in incorrect feedback due to the respondent's subjective misperception of the current situation. Most importantly, the data provided by the two methods are more artificial evaluation data, and they are unable to provide accurate and detailed industry-related data. In contrast, corporate financial data is the real data disclosed by companies, which can reasonably reflect the business performance and accurately reflect the development of the industry. In addition, according to the synergetics and Haken model, it is the factors within the system that influence the synergy of the system. In analyzing the factors within the supply chain system, the enterprise level is the unit of analysis. Therefore, we adopted judgmental sampling to select the sample and conducted research on the financial statement data of enterprises in the specific industries included in the study. Since the coverage of the advanced manufacturing industry is not clearly defined at present, we followed the practice of Li et al. in 2021 and considered the industries involved in the ten key areas listed in Made in China 2025 as representative industries of the advanced manufacturing industry [57]. According to the information of the manufacturing industry classification in the National Economic Classification (GB/T 4754–2017) 2019 revision, the industries involved in the ten key areas are "computer, communication and other electronic equipment manufacturing," "instrumentation manufacturing," "general equipment manufacturing," "railroad, ship, aerospace, and other transport equipment manufacturing," "automotive manufacturing," "electrical machinery and equipment manufacturing," "special equipment manufacturing," "chemical materials and chemical products manufacturing," "petroleum processing, coking, and nuclear fuel processing industry," "pharmaceutical manufacturing."

System evolution is a long-term process. The factors that affect the system in this process always exist. To align with synergetics principles and to ensure that the factors reflected in the data always exist in the system, we selected enterprises with a long establishment time as the source of sample data. In addition, market capitalization in circulation was used to measure the size of an enterprise's assets. Firms with large asset sizes have greater influence on the industry and its supply chain. Using such enterprises as the data source not only ensures the influence of the factors reflected in the data on the system but also improves the representativeness of the sample enterprises. Hence, we selected manufacturing companies listed before 2012 and in the top ten of the industry in terms of market capitalization as the sample for the study. In this way, the panel data were obtained. The data collection process is as follows: first, the initial samples were selected from the top ten listed companies in the ten industries as described in the previous section based on market capitalization; second, the raw financial data were obtained from RESSET, CSMAR, Wind, and the official websites of the enterprises; third, further processing and adjustment of multiple original financial indicators according to the definition of indicators formed the final indicator data. Some of the data in the above

Table 2. Advanced manufacturing SCC motivation index.

| Motivation | Observation index | Indicator composition | Operational definition | Indicator type |
|---|---|---|---|---|
| Competitive advantages of enterprises | Resource competitiveness | Material and financial resources | Total assets | Positive indicator |
| | | Human resources | Number of employees | Positive indicator |
| | Core competence competitiveness | R&D intensity | R&D expenses/Sales revenue | Positive indicator |
| | | R&D density | R&D expenses/Total assets | Positive indicator |
| | Management competitiveness | Total labor productivity | Operating income/Number of employees | Positive indicator |
| | | Selling expense ratio | Selling expenses/Operating income | Negative indicator |
| | | Working capital turnover ratio | Operating income/Average current assets | Positive indicator |
| Cooperative relationships of enterprises | Product flow efficiency | Inventory turnover days | 360/Inventory turnover ratio | Negative indicator |
| | Capital flow efficiency | Accounts receivable turnover days | 360/Accounts receivable turnover ratio | Negative indicator |
| Interest demands of enterprises | Financial performance | Net profit margin of total assets | Net profit/Total assets | Positive indicator |
| | Market performance | Market share | Enterprise main business income / Business income of the industry | Positive indicator |

database could not be accessed before 2016. Considering the availability of financial data, the study period was determined as 2016–2019 in this study. Ninety-four samples and 4512 observations were finally obtained after excluding the samples with a large number of missing data.

**4.2.2 Indicator system.** To clarify the ability of the three major motivations to influence the self-organization process of the advanced manufacturing supply chain, suitable indicators were selected to measure each of the three factors to ensure the accuracy and comprehensiveness of the measurement [58–61], which constitutes the index system of synergistic motivation of the advanced manufacturing supply chain system, as shown in Table 2. Since sales revenue is not disclosed in the financial statements, main business revenue is used instead of sales revenue [62].

**4.2.3 Data processing.** The different units and magnitudes of the relevant data make it difficult to perform uniform measurements. Therefore, standardization of the raw data is needed:

$$\gamma_{ij} = \begin{cases} \dfrac{[x_{ij} - \min(x_{ij})]}{[\max(x_{ij}) - \min(x_{ij})]} \times 0.9 + 0.1, & \text{when } x_{ij} \text{ is a positive indicator} \\ \dfrac{[\max(x_{ij}) - x_{ij}]}{[\max(x_{ij}) - \min(x_{ij})]} \times 0.9 + 0.1, & \text{when } x_{ij} \text{ is a negative indicator} \end{cases} \tag{8}$$

$x_{ij}$ is the original indicator value calculated from the financial data of listed companies. $\gamma_{ij}$ is the standardized result of the indicator value. $i$ indicates the manufacturing company ($i = 1,...,94$), and $j$ is the indicator composition of each motivation ($j = 1,...,12$). In addition, considering the correlation characteristics between the data, the entropy weighting method

was used to obtain the indicator weights.

$$p_{ij} = \frac{\gamma_{ij}}{\sum_{i=1}^{m} \gamma_{ij}} \tag{9}$$

$$H_j = \sum_{i=1}^{m} H_{ij} = \sum_{i=1}^{m} \left( -\frac{p_{ij} \ln p_{ij}}{\ln m} \right) \tag{10}$$

$$\omega_{ij} = \frac{1 - H_i}{\sum_{i=1}^{m} (1 - H_i)} \tag{11}$$

The entropy weighting method refers to assigning corresponding weights to each observation based on the measurement results of the information contained in each observation. This method can help avoid the subjective arbitrariness of indicator assignment. $H_{ij}$ is the information entropy of each indicator value. $\omega_{ij}$ denotes the weight of each indicator. The number of enterprises is $m$, and its value is equal to 94.

## 5. Analysis of results

### 5.1 Influencing factor and relationship mechanism identification

The data analysis we used was a standard regression analysis. The specific data analysis steps for influence factor identification were as follows: based on the discrete motion Eqs (3) and (4), this study combined corporate competitive advantages, corporate interest claims, and corporate partnerships, respectively. A set of six types of motion equations (They correspond to the 6 hypotheses, respectively) was constructed through the combination. According to the time-series relationship between the independent and dependent variables of the Haren model, each type of motion equation set in our study period corresponds to three time periods, so a total of 18 pairs of motion equation sets were obtained. Next, after standardizing the data and obtaining the indicator weights using the entropy weighting method, the resulting weights were used to calculate the values of these three types of drivers for each year. The results of these values were imported into SPSS 26.0 software. The last composite variable of each movement equation was defined by using the Convert-Calculate Variables operation in SPSS 26.0 software. In addition, regression analyses were performed for each of the 18 pairs of motion equations one by one. With the regression analysis results of the SPSS 26.0 software, the coefficients of the motion equations could be solved and the model assumptions could be discriminated. After these operations, the primary influences (sequential parameters) and secondary influences of SCC could be identified from the three synergistic motives, and the results of the self-organization process 1 of advanced manufacturing SCC could be clarified.

The steps of sequential parameter identification are as follows: (1) Assume that one motive is a slow variable $q_1$ and the other is a fast decaying variable $q_2$. Construct the corresponding equations of motion accordingly. (2) Use the significance level of the year-by-year regression results of SPSS 26.0 software to determine whether the equation of motion is valid or not. (3) According to the coefficients of the equation of motion, check whether the parameters of the equation of motion meet the adiabatic approximation condition. (4) Determine whether the model assumptions are valid according to steps (2) and (3). In the process of data analysis, $Z$, $S$, and $W$ were used to represent competitive advantages of enterprises, interest demands of enterprises, and cooperative relationships of enterprises, respectively. Due to the limitation of space, the specific steps and all the data are not listed. The model assumptions and their results are shown in Table 3.

**Table 3. Model assumptions and results.**

| Model assumptions | Period | Equation of motion | Model discriminant |
|---|---|---|---|
| $q_1 = Z$<br>$q_2 = S$ | 2016–2017 | $Z_{2017} = (1 - a_1)Z_{2016} - b_1 Z_{2016} S_{2016}(***)$<br><br>$S_{2017} = (1 - a_2)S_{2016} + b_2 Z_{2016}^2(***)$<br><br>$a_1 = 0.014(***),\ a_2 = -0.019(***)$<br><br>$b_1 = -0.013,\ b_2 = -0.027$ | The equations of motion do not hold. Model assumptions do not hold. |
| | 2017–2018 | $Z_{2018} = (1 - a_1)Z_{2017} - b_1 Z_{2017} S_{2017}(***)$<br><br>$S_{2018} = (1 - a_2)S_{2017} + b_2 Z_{2017}^2(***)$<br><br>$a_1 = 0.027(***),\ a_2 = 0.151(***)$<br><br>$b_1 = -0.030(*),\ b_2 = 0.150(***)$ | The equations of motion hold; the parameters meet the adiabatic approximation. The model assumptions hold. Z is the sequential parameter and S is the fast variable. |
| | 2018–2019 | $Z_{2019} = (1 - a_1)Z_{2018} - b_1 Z_{2018} S_{2018}(***)$<br><br>$S_{2019} = (1 - a_2)S_{2018} + b_2 Z_{2018}^2(***)$<br><br>$a_1 = -0.005(***),\ a_2 = -0.006(***)$<br><br>$b_1 = 0.013,\ b_2 = -0.008$ | The equations of motion do not hold. Model assumptions do not hold. |
| $q_1 = Z$<br>$q_2 = W$ | 2016–2017 | $Z_{2017} = (1 - a_1)Z_{2016} - b_1 Z_{2016} W_{2016}(***)$<br><br>$W_{2017} = (1 - a_2)W_{2016} + b_2 Z_{2016}^2(***)$<br><br>$a_1 = 0.173(***),\ a_2 = 0.048(***)$<br><br>$b_1 = -0.172(**),\ b_2 = 0.046$ | The equations of motion do not hold. Model assumptions do not hold. |
| | 2017–2018 | $Z_{2018} = (1 - a_1)Z_{2017} - b_1 Z_{2017} W_{2017}(***)$<br><br>$W_{2018} = (1 - a_2)W_{2017} + b_2 Z_{2017}^2(***)$<br><br>$a_1 = -0.024(***),\ a_2 = 0.005(***)$<br><br>$b_1 = 0.028,\ b_2 = 0.004$ | The equations of motion do not hold. Model assumptions do not hold. |
| | 2018–2019 | $Z_{2019} = (1 - a_1)Z_{2018} - b_1 Z_{2018} W_{2018}(***)$<br><br>$W_{2019} = (1 - a_2)W_{2018} + b_2 Z_{2018}^2(***)$<br><br>$a_1 = -0.258(***),\ a_2 = 0.004(***)$<br><br>$b_1 = 0.265(***),\ b_2 = 0.002$ | The equations of motion do not hold. Model assumptions do not hold. |
| $q_1 = S$<br>$q_2 = W$ | 2016–2017 | $S_{2017} = (1 - a_1)S_{2016} - b_1 S_{2016} W_{2016}(***)$<br><br>$W_{2017} = (1 - a_2)W_{2016} + b_2 S_{2016}^2(***)$<br><br>$a_1 = -0.210(***),\ a_2 = 0.016(***)$<br><br>$b_1 = 0.219(***),\ b_2 = 0.013$ | The equations of motion do not hold. Model assumptions do not hold. |
| | 2017–2018 | $S_{2018} = (1 - a_1)S_{2017} - b_1 S_{2017} W_{2017}(***)$<br><br>$W_{2018} = (1 - a_2)W_{2017} + b_2 S_{2017}^2(***)$<br><br>$a_1 = 0.339(***),\ a_2 = 0.003(***)$<br><br>$b_1 = -0.330(**),\ b_2 = 0.001$ | The equations of motion do not hold. Model assumptions do not hold. |
| | 2018–2019 | $S_{2019} = (1 - a_1)S_{2018} - b_1 S_{2018} W_{2018}(***)$<br><br>$W_{2019} = (1 - a_2)W_{2018} + b_2 S_{2018}^2(***)$<br><br>$a_1 = -0.054(***),\ a_2 = 0.003(***)$<br><br>$b_1 = 0.057,\ b_2 = 0.000$ | The equations of motion do not hold. Model assumptions do not hold. |

*(Continued)*

**Table 3.** (Continued)

| Model assumptions | Period | Equation of motion | Model discriminant |
|---|---|---|---|
| $q_1 = S$ $q_2 = Z$ | 2016–2017 | $S_{2017} = (1 - a_1)S_{2016} - b_1 S_{2016} Z_{2016}(***)$ $Z_{2017} = (1 - a_2)Z_{2016} + b_2 S^2_{2016}(***)$ $a_1 = -0.108(***),\ a_2 = 0.011(***)$ $b_1 = 0.128(***),\ b_2 = 0.012$ | The equations of motion do not hold. Model assumptions do not hold. |
| | 2017–2018 | $S_{2018} = (1 - a_1)S_{2017} - b_1 S_{2017} Z_{2017}(***)$ $Z_{2018} = (1 - a_2)Z_{2017} + b_2 S^2_{2017}(***)$ $a_1 = 0.240(***),\ a_2 = 0.020(***)$ $b_1 = 0.256(***),\ b_2 = 0.026(*)$ | The equations of motion hold. The adiabatic approximation condition is not satisfied. Model assumptions do not hold. |
| | 2018–2019 | $S_{2019} = (1 - a_1)S_{2018} - b_1 S_{2018} Z_{2018}(***)$ $Z_{2019} = (1 - a_2)Z_{2018} + b_2 S^2_{2018}(***)$ $a_1 = -0.030(***),\ a_2 = 0.000(***)$ $b_1 = 0.036(**),\ b_2 = -0.008$ | The equations of motion do not hold. Model assumptions do not hold. |
| $q_1 = W$ $q_2 = Z$ | 2016–2017 | $W_{2017} = (1 - a_1)W_{2016} - b_1 W_{2016} Z_{2016}(***)$ $Z_{2017} = (1 - a_2)Z_{2016} + b_2 W^2_{2016}(***)$ $a_1 = 0.042(***),\ a_2 = 0.106(***)$ $b_1 = -0.032,\ b_2 = 0.109(***)$ | The equations of motion do not hold. Model assumptions do not hold. |
| | 2017–2018 | $W_{2018} = (1 - a_1)W_{2017} - b_1 W_{2017} Z_{2017}(***)$ $Z_{2018} = (1 - a_2)Z_{2017} + b_2 W^2_{2017}(***)$ $a_1 = -0.001(***),\ a_2 = -0.023(***)$ $b_1 = 0.004,\ b_2 = -0.028$ | The equations of motion do not hold. Model assumptions do not hold. |
| | 2018–2019 | $W_{2019} = (1 - a_1)W_{2018} - b_1 W_{2018} Z_{2018}(***)$ $Z_{2019} = (1 - a_2)Z_{2018} + b_2 W^2_{2018}(***)$ $a_1 = 0.013(***),\ a_2 = 0.025(***)$ $b_1 = -0.010,\ b_2 = 0.022$ | The equations of motion do not hold. Model assumptions do not hold. |
| $q_1 = W$ $q_2 = S$ | 2016–2017 | $W_{2017} = (1 - a_1)W_{2016} - b_1 W_{2016} S_{2016}(***)$ $S_{2017} = (1 - a_2)S_{2016} + b_2 W^2_{2016}(***)$ $a_1 = 0.022(***),\ a_2 = 0.090(***)$ $b_1 = -0.013,\ b_2 = 0.096(***)$ | The equations of motion do not hold. Model assumptions do not hold. |
| | 2017–2018 | $W_{2018} = (1 - a_1)W_{2017} - b_1 W_{2017} S_{2017}(***)$ $S_{2018} = (1 - a_2)S_{2017} + b_2 W^2_{2017}(***)$ $a_1 = 0.001(***),\ a_2 = -0.189(***)$ $b_1 = 0.002,\ b_2 = -0.223(***)$ | The equations of motion do not hold. Model assumptions do not hold. |
| | 2018–2019 | $W_{2019} = (1 - a_1)W_{2018} - b_1 W_{2018} S_{2018}(***)$ $S_{2019} = (1 - a_2)S_{2018} + b_2 W^2_{2018}(***)$ $a_1 = 0.004(***),\ a_2 = 0.041(***)$ $b_1 = -0.001,\ b_2 = 0.045(***)$ | The equations of motion do not hold. Model assumptions do not hold. |

Note: ***, **, and * represent significant at 1%, 5%, and 10% statistical levels, respectively.

For the advanced manufacturing supply chain, the empirical results of the above model assumptions are summarized in Table 4 for a more intuitive understanding of the point-in-time and influencing factors of the phase change of the system.

**Table 4. Model assumptions and results.**

| Sequential parameter | Fast decaying variable | Whether the model is established | Model establishment period |
|---|---|---|---|
| Competitive advantages of enterprises | Interest demands of enterprises | Yes | 2017–2018 |
| | Cooperative relationships of enterprises | No | None |
| Interest demands of enterprises | Competitive advantages of enterprises | No | |
| | Cooperative relationships of enterprises | No | |
| Cooperative relationships of enterprises | Competitive advantages of enterprises | No | |
| | Interest demands of enterprises | No | |

As can be seen from Table 4, in 2017–2018, the advanced manufacturing supply chain underwent a phase change, which is a critical abrupt change from the old stable state to the new state. At the critical point of supply chain collaborative evolution, the factors that affect the collaborative evolution of each part of the system are competitive advantages of enterprises and interest demands of enterprises. The sequential parameter is the competitive advantages of enterprises, which dominates the change of interest demands and further dominates the collaborative evolution of the advanced manufacturing supply chain. H1 was accepted. This is likely due to *Guidance on Deepening the "Internet + Advanced Manufacturing" and Development of Industrial Internet* issued by the State Council in November 2017. Since then, the "Industrial Internet" has become an important force in the digitalization and intelligent transformation of the Chinese manufacturing industry. It has been widely used in advanced manufacturing industries such as petroleum and petrochemical, iron, and steel metallurgy, and so forth. New digital technologies are widely used in the green and intelligent transformation of the manufacturing industry [63]. Advanced manufacturing industry gains development. For example, the application of IoT in 3D printing has significantly improved productivity, which allows 3D printing technology with personalized production to meet the mass customization capabilities required by Industry 4.0 [64]. This allows manufacturing companies to achieve modern and advanced manufacturing processes that are efficient, low cost, high quality, flexible, and automated, so they can produce products with Industry 4.0 standards [65]. As a result, new modes and new business models, such as networked collaboration, service-oriented manufacturing, and personalized customization, have emerged in the manufacturing industry, helping advanced manufacturing enterprises to improve their quality and efficiency. In this process, advanced manufacturing companies continuously give rise to new competitive advantages, which, in turn, satisfy their interest demands. Therefore, the model hypothesis formed by interest demands of enterprises as the sequential parameter and competitive advantages of enterprises as the fast decaying variable cannot be established. H3 was rejected. In addition, H2, H4, H5, and H6 were rejected. The model hypotheses formed by cooperative relationships of enterprises were not valid in each period, which indicates that cooperative relationships were not the main role parameter influencing the collaborative evolution in the advanced manufacturing supply chain system. Cooperative relationships use communication as a starting factor. Commitment and cooperation can be generated after building a trusting relationship between partners. Cooperative relationships have a positive impact on synergy. However, the improvement of partnership and SCC do not necessarily go hand in hand. This is because most domestic manufacturing companies recognize the importance of establishing good supply chain partnerships and are willing to develop long-term relationships with each other, but they do not clearly understand the specific value of developing such relationships. That is, they do not simultaneously recognize the importance of synergy [23]. Therefore, a good supply chain partnership does not necessarily trigger synergistic effects.

The parameters of the equations of motion in Table 2 show the interrelationships between the fast and slow variables and their effects on the system evolution. First, the slow variable (competitive advantages of enterprises) and the fast decaying variable (interest demands of enterprises) promote each other and complement each other. $b_1$ is -0.030, which indicates that interest demands of enterprises play a boosting role in competitive advantages of enterprises. $b_2$ is 0.150, which suggests that competitive advantages of enterprises promote the growth of interest demands of enterprises. Here is an example of an advanced manufacturing company for an explanation. After achieving stable operation, companies with a certain competitive advantage pay attention to the growth of corporate profits. They will improve profits from scale and cost. They further develop downstream customers on one hand. On the other hand, they value getting materials from upstream suppliers at a lower cost. Suppliers with unique capabilities and good performance are likely to gain the trust of manufacturing companies. Through the integration of advantages, advanced manufacturing companies and suppliers benefit together, which is conducive to both parties' improving corporate performance. Second, there is a self-positive feedback mechanism for both the slow variable (competitive advantages of enterprises) and the fast decaying variable (interest demands of enterprises). $a_1 = 0.027<1$, which shows that more members in the supply chain have unique advantages when the level of SCC in the advanced manufacturing industry increases. At this time, the complementary advantages of supply chain members based on synergistic effects, such as technological progress generated by collaborative innovation, are enriched continuously. $a_2 = 0.151<1$, which means that the revenue target of enterprises gradually increases when the level of SCC in advanced manufacturing continues to improve. This is the purpose of enterprises promoting SCC.

## 5.2 Analysis of the effect of main influencing factors

Based on the identification of the influencing factors of SCC, the realization mechanism of the self-organization process 2 of advanced manufacturing SCC can be clarified, that is, the competitive advantages control the advanced manufacturing SCC and dominate the interest claims to participate in it as well. Subsequently, the advanced manufacturing supply chain is in a process of forming a new orderly structure. In this process, competitive advantages of enterprises that dominated system gradually move from disorder to order and then to high order. Therefore, the root of SCC in the advanced manufacturing industry lies in competitive advantages of enterprises.

Competition of advantages between supply chain partners include two situations. The first is the differential advantage. It means that both parties have unique key resources and capabilities that cannot be replaced by each other. In this case, synergy is to achieve long-term stable cooperation by maximizing the benefits of both parties. The second is comparative advantage. Comparative advantage means that one party has better resources or capabilities than its partner in a certain aspect. The weaker party enhances its competitiveness through the superior resources or capabilities of its partner. Therefore, the weaker party has the risk of being replaced. At this time, the partnership is unstable.

According to the regression results of the equation of motion in Table 2 and Eq (7), the image of the potential function is drawn, as shown in Fig 3. In the process of collaborative evolution of the supply chain in the advanced manufacturing industry, competitive advantages of enterprises always exist. Therefore, only $Z>0$ is considered here. The process in which the sequence parameter of phase change point continuously changes from zero to non-zero value is the process in which the system changes from disorder to order. The sequence parameter and potential energy of point A are both 0, which indicates that the system tends to disorder.

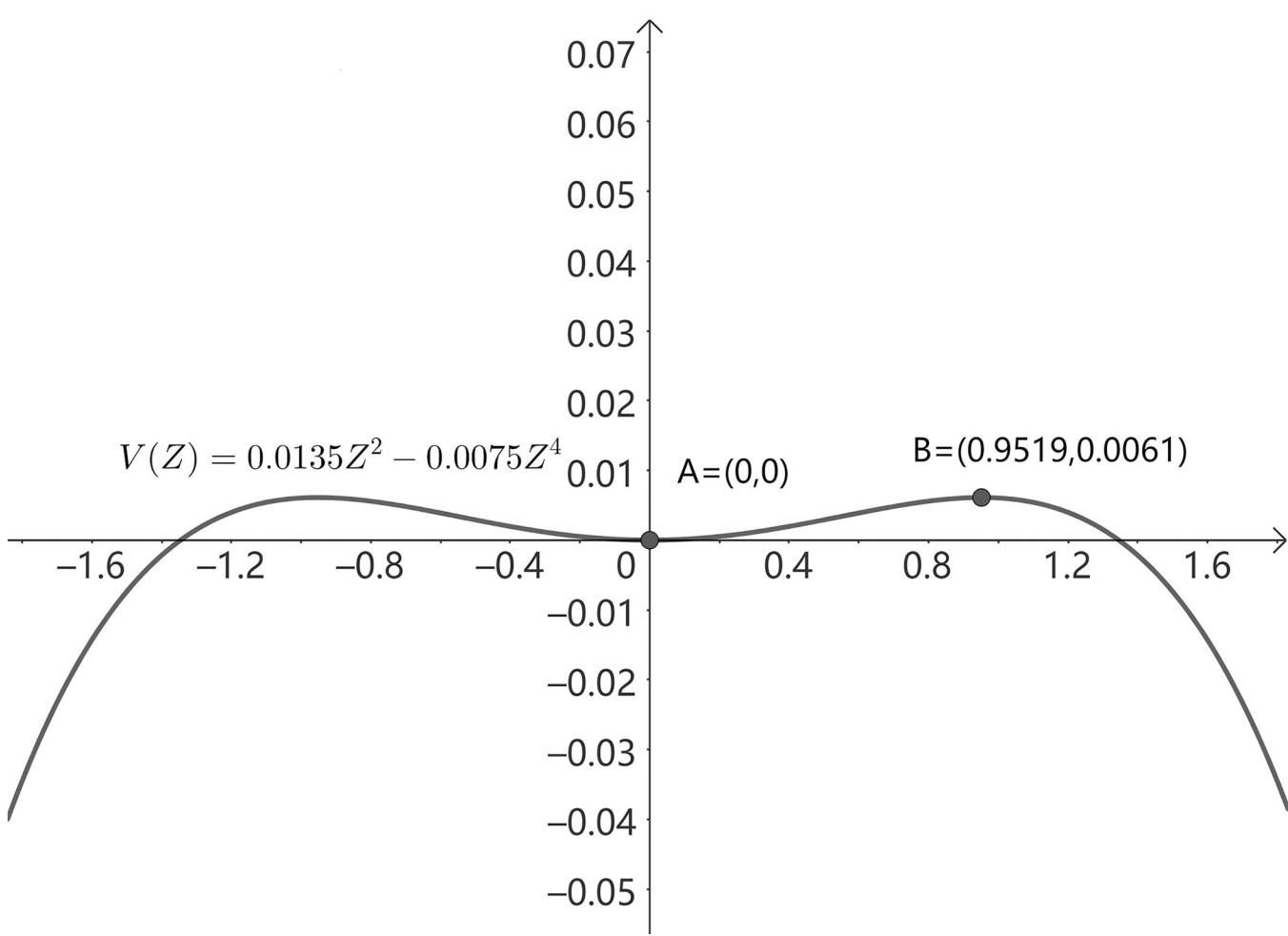

**Fig 3. Potential function images of phase change points in advanced manufacturing supply chains.**

When the system motion state tends to A, the value of the competitive advantages index of the advanced manufacturing enterprise is close to 0. At this time, competitive advantages of enterprises are relatively lacking, so enterprises cooperate with upstream and downstream enterprises based on comparative advantage. The sequential parameter at point B has a non-zero value and the potential energy at this point is the highest. The system is in a high level of orderly state at point B. When the system movement state tends to B, the value of the competitive advantages index of the advanced manufacturing enterprise is close to 0.95. At this time, the enterprise has a strong competitive advantage and completes cooperation with supply chain partners based on differential advantage. In this case, the members of the supply chain all become essential members of a stable partnership with their unique resources and capabilities, and their strengths are comparable but mutually nourishing. As the upstream and downstream enterprises form a collaborative relationship based on their differential advantages, each enterprise obtains the maximum synergistic effect based on the synergy and spillover effect of their differential advantages. Accordingly, the whole supply chain also obtains the maximum synergistic effect and finally reaches the "supply chain synergistic competitive equilibrium state."

## 6. Discussion

The results of our study provide new insights into the development of SCC in advanced manufacturing industries in the following ways:

There are no empirical studies exploring the influencing factors of SCC in terms of the nature of synergy, and only a few studies exist to analyze the feedback capability of synergistic influence. On the one hand, the validation results of the Haken model show that, in the synergy dynamics framework constructed based on the theory of comparative advantage and synergetics, the competitive advantage of enterprises is the sequential parameter in the process of SCC in China's advanced manufacturing industry and is the main and fundamental cause of SCC. This result is consistent with the results of past studies on competitive advantage and the level of supply chain modernization; for example, the upgrading of advanced manufacturing supply chains requires that the improvement of firms' competitive advantages be integrated into the core of supply chain optimization [66, 67], which is because the cultivation of advantages based on innovation and its application can promote the improvement of supply chain efficiency and effectiveness and the further improvement of supply chain governance effectiveness [68]. This result also proves that, considering that the integration of advantages at scale contributes to industrial upgrading [69], so also the substantial technological advantage gained from technological catch-up in advanced manufacturing helps to break through the technology bottleneck of the industry and achieve leapfrogging in the supply chain of autonomous industries [70]. These may reflect the fact that the competitive advantage of advanced manufacturing firms and their supply chain members that are important "engines" of China's economy and active areas of innovation has received attention and encouragement from many parties inside and outside the supply chain due to the change of China's economic development to being innovation-driven as a result of, for example, the weakening of China's cost advantage. On the other hand, the study shows that the impact of competitive advantage on the system has a positive feedback capability on the construction of competitive advantage itself. This is consistent with the effect of competitive advantage on SCC in past studies [14, 36]. It is not difficult to infer that this positive feedback forms a virtuous cycle between SCC and competitive advantages, which may contribute to the dynamic competitive advantages of Chinese advanced manufacturing firms and their supply chains. In this regard, Chinese advanced manufacturing firms and supply chain partners may realize substantial and sustained innovation benefits based on relationship governance in order to achieve this goal.

There is no literature that can distinguish the interactive forces of competitive advantages and benefit claims, and there is a gap in the research on the ranking of the level of influence of the two on SCC. This study shows that, in advanced manufacturing supply chains, the positive impact of benefit claims on competitive advantages is smaller than the positive impact of competitive advantages on benefit claims. The result shows the phenotype of firms tapping benefits with advantages, as mentioned in previous studies. For example, in a supply chain, the dynamic competitiveness built by firms collaborating fully is for maximum profit [54]. The positive feedback capability of the synergistic impact of competitive advantages and benefit claims ensures that "the improvement of the level of SCC contributes to the construction of firms' competitive advantage and fuels their benefit claims." However, when faced with the choice between building competitive advantage and benefit claims after improving the level of synergy, the willingness of firms to build competitive advantages may be reduced by benefit claims, which leads to short-term behavior of "benefit first." This is not conducive to long-term sustainable development and long-term cooperation in the supply chain. Fortunately, our results show that the impact of interest claims on competitive advantage is lower than the impact of competitive advantages on interest claims. This reflects the fact that the Chinese

manufacturing supply chain members may avoid the above-mentioned short-term profit-seeking behaviors in the face of competitive advantages construction and benefits, thus maintaining the stability of the supply chain. In addition, as for influencing supply chains collaboration in Chinese advanced manufacturing industries, our results show that the positive influence of interest claims is subordinated to the positive influence of competitive advantages. This result validates that "competitive advantages is the ordinal parameter" and also validates the influence of interests on manufacturing SCC in previous studies. For example, supply chain cooperation is for mutual benefit [29]. This reveals that firms in China's advanced manufacturing supply chains are likely not to use immediate benefits as the first trade-off when faced with SCC decisions, but rather, the primary consideration is more likely to be how the decision can gain or enrich competitive advantages.

The above new insights establish a mechanism and comparison of the relationship between competitive advantages, interest demands, and SCC. Therefore, these new insights can also be used to address the inquiry of the basis of SCC governance in China's advanced manufacturing industry. In addition, they can be used to explain the validity of resource- and capability-based supply chain governance measures. Finally, these new insights are applicable to the study of Chinese advanced manufacturing firms' strategies in the context of SCC.

## 7. Conclusion

In this paper, the synergistic motives developed from the theory of comparative advantage and synergetics are applied to the advanced manufacturing supply chain system, and the Haken model is used to find out the factors influencing the escalation of the level of synergy in the advanced manufacturing supply chain. Through these, this study draws several conclusions.

First, in 2017–2018, the advanced manufacturing supply chain system in China underwent a phase change. The fundamental influencing factor (sequential covariate) was the firms' competitive advantages. The secondary influencing factor was the firms' interest claims. This is consistent with the relationship between interest, competitive advantage, and manufacturing SCC provided in the literature review. However, our findings reinforce the distinction between the two influencing capabilities. It is possible to see that members of advanced manufacturing supply chains will increasingly focus on competitive advantages in the coming years relative to profit goals. To build a strong competitive advantage and achieve an upgrade in the level of synergy across the supply chain, they will not only enhance activities related to technological innovation, but also continuously improve their production management practices. Second, there is a positive interaction between competitive advantages and interest demands. Both of them have positive interaction with SCC. This reveals the necessity and relevance of competitive advantages and SCC. Faced with the negative impact of increasing global economic downward pressure on competitive ability and business performance, Chinese advanced manufacturing industries will likely commit to collaborative supply chain governance in the coming period as a way to maintain or even improve advantages, such as global market share, supply chain resilience, and gaining greater profits. In addition, they are likely to further strengthen collaborative governance with improved capabilities and profits. Finally, when firms cooperate based on differential advantage (i.e., the order parameter tends to 0.95), the supply chain system is in the advanced orderly stage of the collaborative competitive equilibrium. Our research shows that the combination of superior firms in the supply chain is the highest stage of supply chain upgrading. Accordingly, it is not difficult to guess that the competitive advantage of the supply chain will be better when there are more cooperative links based on differential advantages in the Chinese advanced manufacturing supply chain. In this case, this greater competitive effect of SCC can be generated.

 

## 7.1 Theoretical significance

This research contributes to the development of understanding related to manufacturing SCC.

Our theoretical contribution is the development of a triadic framework of "competitive advantages-interest demands-cooperative relationships" that is widely applicable to the synergistic dynamics of socio-economic systems. On the one hand, this framework is the first to link the theory of comparative advantage and synergetics. On the other hand, it is based on the nature of synergy, which ensures that the derived influencing factors satisfy the logical conditions that govern the system "behind the scenes" of the sequential parameters. This is the key difference from the previous literature in the field of drivers of SCC.

Equally important, the study contributes by showing that the positive influence of interest claims on competitive advantages is less than the positive influence of competitive advantages on interest claims and that the influence of interest claims on SCC in advanced manufacturing is dominated by competitive advantage, as opposed to the superficiality of economic phenomena and gaps in previous research. This comparative analysis enriches the previous one-way analysis of the relationship between advantage and interest and the influence they have on SCC. This highlights the overwhelming role of competitive advantage in upgrading the level of SCC in advanced manufacturing. For the first time, direction is provided for advanced manufacturing practitioners to improve SCC, and economic phenomena in the field of collaboration that have not been analyzed in the literature are explained. In recent years, advanced manufacturing enterprises in many cities in China have been building on their own economic goals, advantages, and needs and have been improving their SCC to become the leading global advanced manufacturing enterprises and to move up the global industrial chain. However, when interests and competitive advantages drive advanced manufacturing SCC, it is not possible to perceive or observe which of the two factors plays the major role. This has not been addressed in previous theoretical studies. This study identifies this unidentified and unexamined issue. In addition, the intensity of R&D investment in China's manufacturing industry increased from 0.85% in 2012 to 1.54% in 2021, and the share of new product revenue in business revenue of manufacturing companies with a certain scale increased from 11.9% in 2012 to 22.4% in 2021 [71]. The proportion of R&D investment in advanced manufacturing industries is higher than the overall level [72]. At the macro-industry level, we can observe that the total R&D investment in the industry is increasing as a percentage of the total operating revenue. At the micro-enterprise level, we can see that, to gain a competitive advantage, many high-tech enterprises are investing an increasing proportion of their total revenue in research. For example, Huawei's research and development investment for 2019 was 131.7 billion yuan, and the proportion of R&D investment to annual revenue was 15.3% [73]. These economic phenomena show the relationship between the competitive advantages generated by innovation and the profit of the firm, but they also show that the competitive advantages generated at great cost do not immediately result in an equal amount of profit. These phenomena cannot show whether competitive advantage has a greater influence on benefit claims or if benefit claims have a greater influence on competitive advantage. Thus, this study fills the explanatory gap of the economic phenomenon of SCC in the advanced manufacturing industry through the comparison of impacts.

## 7.2 Management inspiration

This paper provides management insights for senior decision makers and practitioners in advanced manufacturing companies.

In terms of supply chain management, managers of advanced manufacturing companies need to focus on innovation and the cultivation of competitive advantages. The research

shows that competitive advantage is the fundamental factor contributing to SCC. The shaping of competitive advantage depends to a large extent on innovation. Collaboration between firms creates the conditions for both parties to integrate innovative knowledge and resources. The benefits of this process lead to a closer and more sustainable collaboration, which, in turn, leads to an increased level of SCC. Therefore, in order to promote SCC, top-level decision makers of enterprises need to incorporate the concept of innovation into their daily management activities and cooperation strategies with supply chain members according to the new trend of innovation. On the one hand, top managers of enterprises should encourage employees to innovate in terms of talent training and achievement transformation. They can make full use of Internet technology to organize collaborative innovation or technical training for staff in production and R&D departments between supply chain partners. Technical exchanges and innovations can be encouraged to solve the shortcomings of China's advanced manufacturing supply chain in terms of key components and raw materials, to realize the matching of new technologies and parts, and to improve the collaboration ability of the whole supply chain in terms of product innovation standards. On the other hand, senior decision makers also need to cooperate and learn from partners in terms of innovative management processes, organizational structures, and management models based on information technology, automation, and intelligence. By doing so, they can create operational advantages and improve business processes to better serve the development of new processes, technologies, and products, which can maximize the efficiency of collaboration in all parts of the supply chain. For example, manufacturers can develop new services or business models based on end-consumer demand and leverage the connectivity of cloud technology to allow relevant partners to align processes with visible consumer demand.

Sourcing and sales managers in advanced manufacturing companies need to select partners with specific strengths in their industries. Our research shows that a differentiated advantage mix of partner strengths can help maximize synergies. When a partner is selected that has a competitive advantage in an area that is not available in the same industry, it is possible to gain benefits from working with them that are not available with other options. Specifically, purchasing managers and sales managers need to clarify whether the measures the partner will take will benefit the company in terms of efficiency, cost, quality, responsiveness, and so forth based on the company's long-term development strategy, core competencies, and partner evaluation results. On the one hand, the most competent companies should be selected as long-term strategic partners among companies that have business relationships. On the other hand, as the development of supply partnerships is subject to changes in market demand, purchasing and sales managers need to change their requirements for partners or reselect partners in a timely manner according to the actual needs of the company. This will enable the re-matching of the supply chain differential advantage portfolio.

### 7.3 Limitations and future research

There are some limitations in this study. First, the findings only show and explain that, when the level of SCC in advanced manufacturing is upgraded, benefit claims are the channel through which competitive advantage exerts its influence. Further research could focus on analyzing whether interest claims are fully or partially mediated. Further research can compare the magnitude of these two roles and thus provide firms with more dimensions of SCC strategies. Second, our research results are only applicable to Chinese advanced manufacturing supply chains, and there are only methodological implications for other countries' advanced manufacturing supply chain systems. Further research can analyze the influencing factors of SCC in advanced manufacturing industries in other countries, compare the differences in

research results between countries and analyze the reasons for the differences. This future research can then provide guiding suggestions for the development of advanced manufacturing industries in each country. Third, the study only shows that the fundamental influencing factor of supply chain upgrading in China's advanced manufacturing industry in the recent period is the competitive advantages of enterprises and that the secondary influencing factor is the interest claims. Subsequent studies can focus on the change of influencing factors and the prediction of change time.

## Supporting information

**S1 File. Raw data.**
(XLSX)

**S2 File. The result of processing raw data by entropy weight method.**
(XLSX)

## Author Contributions

**Conceptualization:** Haohua Liu, Lin Lin, Xiuling Chen, Jianwei Ren.

**Data curation:** Lin Lin, Xiuling Chen.

**Formal analysis:** Lin Lin, Yang Cheng, Xiuling Chen.

**Funding acquisition:** Haohua Liu, Xiuling Chen, Jianwei Ren.

**Investigation:** Lin Lin, Jianwei Ren.

**Methodology:** Lin Lin, Xiuling Chen.

**Project administration:** Lin Lin.

**Software:** Lin Lin.

**Supervision:** Haohua Liu, Yang Cheng, Xiuling Chen, Jianwei Ren.

**Validation:** Haohua Liu, Lin Lin, Xiuling Chen.

**Visualization:** Lin Lin.

**Writing – original draft:** Haohua Liu, Lin Lin, Xiuling Chen.

**Writing – review & editing:** Lin Lin, Yang Cheng.

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
