## [Decision Letter · Decision Letter 0]

14 Oct 2022

PONE-D-22-25768

Title -Factors influencing phase change points in the collaborative evolution of advanced manufacturing supply chains

PLOS ONE

Dear Dr. Lin,

Thank you for submitting your manuscript to PLOS ONE. After careful consideration, we feel that it has merit but does not fully meet PLOS ONE’s publication criteria as it currently stands. Therefore, we invite you to submit a revised version of the manuscript that addresses the points raised during the review process.

ACADEMIC EDITOR:

Dear author,

the reviewers provide several insights. In addition, I highlight some aspects: abstract, you can report the main implications of your study within section 1 better explain the novelty of your work improve the literature with recent work Within the text start sentences with authors' names, always give a logical flow between sentences. discuss the different circular alternatives. conclusions report the limitations of your work.

We look forward to receiving your revised manuscript.

Kind regards,

Muhammad Ikram

Academic Editor

PLOS ONE

Journal Requirements:

2. PLOS ONE does not copy edit accepted manuscripts (https://journals.plos.org/plosone/s/criteria-for-publication#loc-5). To that effect, please ensure that your submission is free of typos and grammatical errors.

Reviewers' comments:

Reviewer's Responses to Questions

Comments to the Author

1. Is the manuscript technically sound, and do the data support the conclusions?

Reviewer #1: Yes

Reviewer #2: Yes

2. Has the statistical analysis been performed appropriately and rigorously?

Reviewer #1: Yes

Reviewer #2: Yes

3. Have the authors made all data underlying the findings in their manuscript fully available?

Reviewer #1: Yes

Reviewer #2: Yes

4. Is the manuscript presented in an intelligible fashion and written in standard English?

Reviewer #1: Yes

Reviewer #2: Yes

5. Review Comments to the Author

Reviewer #1: Dear Authors,

Introduction and Literature review section are fall to provide exact novelty, finding, and problem definition. Thus, authors are advised to recreate those section very carefully.

Literature review does not provide a state of the art about the main concepts that are discussed in this paper. The paper does not demonstrate an adequate understanding of the relevant literature in the field. Authors should analyze the findings and research gaps from previous researchers. Refer to the following articles:

Kraude, R., Narayanan, S., & Talluri, S. (2022). Evaluating the performance of supply chain risk mitigation strategies using network data envelopment analysis. European Journal of Operational Research.

Urbaniak, M., Zimon, D., Madzik, P., & Šírová, E. (2022). Risk factors in the assessment of suppliers. Plos one, 17 (8), e0272157.

Jum, L., Zimon, D., & Ikram, M. (2021). A relationship between supply chain practices, environmental sustainability and financial performance: evidence from manufacturing companies in Jordan. Sustainability, 13 (4), 2152.

Consluson section is too long. Divide it into "discussion" and "conclusions" sections. In the Discussion section, the obtained results should be confronted with the opinions contained in the literature on the subject and with the results of the other studies, so that it is clear whether the obtained results confirm or question the current state of knowledge. In the "conclusion" section, on the other hand, focus on the final conclusion, research limitations, implications and future research directions.

Good Luck!

Reviewer #2: The researchers' research topic is very interesting; there are few studies conducted on this subject and approach, however very necessary. The review of the literature is thorough, and the author has given an adequate background about the topic. The findings clearly demonstrate the contribution to the existing literature. Overall, this is a clear, concise, and well-written manuscript. -The title should be changed to reflect more clearly the contribution of the study. Please revise the Conclusion in the abstract to avoid overly casual language.

Overall the paper has the potential to make a significant contribution to the literature. The model, the data, and the analyses are all fine, so all my issues are with things that can be fixed - with some rewriting, this could be a great paper.

-- In the Introduction, you need to connect the state of the art to your paper goals. Please

follow the literature review by a clear and concise state of the art analysis. This should

clearly show the knowledge gaps identified and link them to your paper goals. Please

reason both the novelty and the relevance of your paper goals. Clearly discuss what the

previous studies that you are referring to.What are the Research Gaps/Contributions? Please note that the paper may not be considered further without a clear research gap and novelty of the study.

-In the Introduction, the 2nd paragraph must support a practical example of the problem descriptions. A potential suggestion to the author/s is to support the importance of a title with some practical examples. However, too many things are bundled together, so the logic and results are not very clear or convincing.

-Page11 manuscript, line 229 to 227, authors have made a good linkaged among the literature. Here I would suggest,where author discuss about the innovation capacity, here,I recommend that add the following with innovation. I encourage authors to carefully consider and incorporate the suggested articles for the manuscript quality improvements and justify wher necessary.

Interorganizational collaboration for innovation improvement in manufacturing: The mediating role of social performance, International journal of innovation management, https://doi.org/10.1142/S1363919620500498

The Impact of Relational Governance on Performance Improvement in Export Manufacturing Firms, Journal of Industrial Engineering and Management, JIEM,11(3): 349-370. DOI: http://dx.doi.org/10.3926/jiem.2558

-Literature review:There is no flow in the text. It partly depends on the lack of proofreading but also on the fact that many statements and claims are made without being followed up by a clear and logical discussion.I suggest the authors to add more literature to make the literature strong. In this study, literature part, the conceptualization of each study variable can be added. Additionally, explain the relationship between the variable of the study. Please explains the use of theory to develop this model. Literature Review has the chance to be further improved: it seems that the authors have made the retrospection. However, via the review, what issues should be addressed? What is the current specific knowledge gap? What implication can be referred to? The above questions should be answered. Authors need to propose their study.

-The discussion needs to be more elaborate and the author should also compare it with previous research in this area. What are the main benefits of this study? The same goes for recommendations and future research.

-Page 25, 467- to 475.I like the way author organize the paper, here author discuss about the advanced manufacturing. It is up to the authors, but it will be good, if they consider, and linked advance manufacturing concept with the industry 4.0 and additive manufacturing. Here, I suggest the following article.This article has highlighted the advance manufacturing.

Designing Value Chains for Industry 4.0 and a Circular Economy: A Review of the Literature. Sustainability. 2022; 14(12):7084. https://doi.org/10.3390/su14127084; Transition toward green economy: Technological Innovation's role in the fashion industry, https://doi.org/10.1016/j.cogsc.2022.100657

Methods

The methods and results show good explanation.As authors know that, method sections determine the results. I suggest refining these sections to remove minor errors. Add justification of this study sample. Kindly focus on the following suggestions

1-how the study was design, discuss complete data collection process, and compare with other methodologies.

2-how the study was carried out, here, highlight the detailed sampling procedures,and pilot testing procedures, if any adopted.

3-How data were analyzed.

The conclusion is very weak.Please make sure your conclusions section underscores the scientific value-added of your paper, and/or the applicability of your findings/results. Highlight the novelty of your study. In addition to summarizing the actions taken and results, please strengthen the explanation of their significance. It is recommended to use quantitative reasoning comparing with appropriate benchmarks, especially those stemming from previous

work.

It should also be an extrapolation of the key findings from the

research and not a summary. So, there should be conclusions around the background theory, data theory/analysis and, key outcomes. The authors should have included the following sub-sections within the conclusion section with more details:

-The authors should offer implications for theory and practice separately as discussed above. See suggested articles to revise the implications and offer actionable points for implementing the offered implications.

--I suggest the authors refine the managerial insights based on the findings

The writing of the paper needs a lot of improvement in terms of grammar, spelling, and presentations. The paper needs careful English polishing since many typos and poorly written sentences exist.

- Limitations in the suggested approach should be discussed in the conclusions section. Implications for future research may also be included in the Conclusion at the end.

Some examples are as the following:

* Check the usage of the commas carefully.

The idea of the paper is quite good and convincing,I'm not going to keep identifying problems with the writing, but please do not just fix these examples and leave the rest - find them all and fix them all. I suggest you have someone fluent in English go over the whole paper and check each sentence.Avoid repetitions. I can see several repetitions at different places in this paper. Thorough proofreading is required.

6. PLOS authors have the option to publish the peer review history of their article (what does this mean?). If published, this will include your full peer review and any attached files.

Do you want your identity to be public for this peer review? For information about this choice, including consent withdrawal, please see our Privacy Policy.

Reviewer #1: No

Reviewer #2: No

---

## [Author Response · Author response to Decision Letter 0]

6 Dec 2022

Response to Editor

Comment 1: in abstract, you can report the main implications of your study within section 1. 

Response 1: Thank you very much for your valuable comments. Based on your guidance, we have added the research significance to the abstract. The research significance we mentioned includes theoretical and practical significance. The theoretical significance is a supplement to the literature and a guide for subsequent research based on the novelty of the refreshed study. The practical significance indicates guidance for practitioners and senior managers in the relevant industries. Also, to make the theoretical and practical implications of the study more intuitive, we have added a brief summary of theoretical gaps and practical issues in the beginning part of the abstract. Finally, in order to highlight our research findings more, we have also added further elements of our conclusions in the abstract. Also, we have mentioned the significance of the study again in the introduction section. In order to highlight the revisions, we have marked the revisions in red in the manuscript (marked in red as L16-43). 

Comment 2: better explain the novelty of your work. 

Response 2: We are super grateful to get your guidance. According to your suggestion, we rethink and express our innovation in the introduction as accurately and clearly as possible. In terms of viewpoint innovation, compared with previous studies, we have developed a motivation framework of “competitive advantages — Interest demands — cooperative relationship” based on the theory of comparative advantage and synergetics, which is unprecedentedly consistent with the characteristics of the effects of order parameters on the system. This not only ensures the selection criteria of synergistic influencing factors for the first time, but also connects the theory of comparative advantage and synergetics for the first time. In terms of perspective innovation, we further compare and prioritise the influence of various possible factors on manufacturing SCC, as opposed to the previous validation studies on the existence of various factors on manufacturing SCC. This points to the general direction and the main grip for upgrading the level of SCC in advanced manufacturing industry. To highlight the revisions, we have marked the revisions in red in the manuscript (marked in red as L105-113). 

Comment 3: improve the literature with recent work Within the text start sentences with authors' names, always give a logical flow between sentences.

Response 3: We are very grateful for these suggestions. Based on your suggestions, the literature in the full text has been updated to the last three years. So the literature included in the literature review section is the most recent results. In addition, in the literature review, we have categorized and distilled the literature analyzed. As far as possible, we have included sentences beginning with the author's name after each summarized sentence. This made our content more logical. To highlight the revisions, we have marked them in red in the manuscript. (marked in red as L143-338). 

Comment 4: discuss the different circular alternatives.

Response 4: We appreciate your guidance on the discussion section. We couldn't agree with you more. This significantly expands the problem-solving scope of our study. We have therefore analyzed different circular alternatives to the new insights related to our empirical results in the final paragraph of the Discussion section. In order to highlight the revised content, we marked the revised content in red in the manuscript (L855-860 in red). 

Comment 5: conclusions report the limitations of your work. 

Response 5: We strongly agree with your suggestion in this regard. We also feel the need to add limitations in the conclusion section. The limitations of our study include three main areas. First, the findings only show and explain that, when the level of SCC in advanced manufacturing is upgraded, benefit claims are the channel through which competitive advantage exerts its influence. Further research could focus on analyzing whether interest claims are fully or partially mediated. Further research can compare the magnitude of these two roles and thus provide firms with more dimensions of SCC strategies. Second, our research results are only applicable to Chinese advanced manufacturing supply chains, and there are only methodological implications for other countries’ advanced manufacturing supply chain systems. Further research can analyze the influencing factors of SCC in advanced manufacturing industries in other countries, compare the differences in research results between countries and analyze the reasons for the differences. This future research can then provide guiding suggestions for the development of advanced manufacturing industries in each country. Third, the study only shows that the fundamental influencing factor of supply chain upgrading in China’s advanced manufacturing industry in the recent period is the competitive advantages of enterprises and that the secondary influencing factor is the interest claims. Subsequent studies can focus on the change of influencing factors and the prediction of change time. (marked in red as L977-992). 

 

Response to Reviewers

Reviewer #1:

We are very grateful for your suggestions. Now we have carefully analyzed and revised exactly according to your comments and found these comments are very helpful. We hope this revision can make our paper more acceptable. The revisions were addressed point by point below. 

Comment 1: Introduction and Literature review section are fall to provide exact novelty, finding, and problem definition. Thus, authors are advised to recreate those section very carefully. 

Response 1: We would like to express our sincere thanks to you. Introduction and Literature review section have been revised following your guidance. (marked in red L74-113, L311-338) 

Innovation in introduction：Compared with the current literature, we have achieved a innovation in viewpoint, i.e., we have developed a triadic motivation framework of “competitive advantages - interest demands - cooperative relationships” based on the theory of comparative advantage and synergetics. We also have a perspective innovation, which is reflected in the fact that, unlike the previous validation studies on the existence of various factors influencing manufacturing SCC, we further compare the influence ability of various possible factors and find the priority of the influence ability. In detail, this study compares the influence of competitive advantages and interest demands on the synergy of advanced manufacturing supply chains and also compares the interaction forces of competitive advantages and interest demands.

Finding based on realistic problem in introduction：As we can see, their numerous purposes of laying out SCC influence their layout measures, which, in turn, influence the level of SCC they are able to achieve. Their ability to achieve a higher level of SCC is due to the fact that their layout measures are related to the main influencing factors of SCC. In fact, it is still difficult to bring about the full implementation of high-level SCC. On one hand, factors affecting SCC, such as the level of information technology [5], environmental protection tendency [6], and profit distribution mechanism [7] are spread throughout the supply chain system. This makes it difficult to focus on the main direction of SCC. On the other hand, synergetics do not indicate the specific factors affecting the synergy but only point out that the sequential parameters were significant and root signs before and after the system phase change and guided the fast variables to influence the synergy of the system [8]. So, in the absence of theories indicating the key specific factors for achieving SCC, many advanced manufacturing companies are at their wits’ end when facing the unprecedentedly complex situation. To effectively cope with the problems caused by their own shortcomings, trade frictions, and the COVID-19 pandemic, Chinese advanced manufacturing supply chains need to find and seize the key to realizing SCC.

Problem definition in introduction：Therefore, the purpose of this study corresponds to the following questions:

1. What are the primary and secondary influencing factors of SCC in China’s advanced manufacturing industry, and what are their relationship mechanisms?

2. In terms of their effect, what state of the main influencing factors will lead to the highest level of SCC in China’s advanced manufacturing industry? 

Innovation in literature review：These shortcomings leave room for this study to develop a theoretically grounded influence factor system of SCC that is consistent with the qualities of the covariates that trigger synergistic effects, complete and comprehensive for the first time, and with broad applicability. They also leave room to clarify for the first time the focal direction of SCC level upgrading in the advanced manufacturing industry through the differentiation ability of synergistic influence factors.

Finding based on theoretical research in literature review： As mentioned above, scholars have explored the theoretical aspects of synergetics and influence factors in SCC. These rich research results have provided good references for this paper. However, the literature also suffers from three shortcomings. First, in the literature of influencing factors, no scholars have mentioned interest claims, and many scholars only think at the level of profit. For example, profit is the purpose of SCC and profit win-win is the form of SCC [7,11]. No scholars have proposed a comprehensive and systematic framework of influencing factors of SCC. Therefore, all the influencing factors of advanced manufacturing SCC cannot be determined. Second, some scholars have selected influencing factors according to the operational characteristics of specific research subjects [6,30]. But this approach has the problem of limited scope of reference, and such apparent reasoning is contrary to the underpinning logic of “dominating the system to produce qualitative changes” of sequential parameters. The elements emphasized in the theory of comparative advantage are the same as those emphasized in the synergistic nature of the economic system in the study of synergetics [45,48], but there is no research to point out and use the relationship between the two theories. Therefore, the current way of determining the influence factors of advanced manufacturing SCC does not take into account the requirements of synergetics for the existence of their influence capabilities. Third, although researchers have suggested that corporate interests and the manifestations of cooperative relationships interact [36,39-41] and can influence manufacturing SCC separately [7,9,34], no literature has yet distinguished and analyzed the primary and secondary effects of interests, competitive advantage, and cooperative relationships on SCC. The mutual influence among competitive advantage, interest claims, and cooperative relationships is also not clearly ranked by the literature. Therefore, the key to upgrading the level of SCC in the advanced manufacturing industry cannot be determined.

Problem definition in literature review：The relevance of the research question is then raised in the process of identifying the research gap. In the text it is mainly manifested in these sentences in the research gap section of 2.4. We have attached them below.

“Therefore, all the influencing factors of advanced manufacturing SCC cannot be determined.”

“Therefore, the current way of determining the influence factors of advanced manufacturing SCC does not take into account the requirements of synergetics for the existence of their influence capabilities. ”

“Therefore, the key to upgrading the level of SCC in the advanced manufacturing industry cannot be determined.”

Comment 2: Literature review does not provide a state of the art about the main concepts that are discussed in this paper. The paper does not demonstrate an adequate understanding of the relevant literature in the field. Authors should analyze the findings and research gaps from previous researchers. 

Response 2: Thank you for your careful guidance. After receiving your valuable comments, we have carefully read the three articles you provided us. From the three articles you presented, we learned that a literature review needs to focus on what is directly related to the research question, i.e., a literature review exists to draw out the research gaps corresponding to the research question by reviewing and sorting through the articles related to the research question. Subsequently, in order to ensure our full understanding of the literature related to this research area, we have reconsidered your comments and strongly agree with your suggestion to “provide a state of the art about the main concepts that are discussed in this paper”. Because we found from the three articles you provided that the articles related to the research question can be identified by examining the main concepts that are discussed. As a result, we have made a best effort to improve the literature review. The improvement idea is as follows: First, our first research question is what are the main and secondary influencing factors and their relationship mechanisms of SCC in Chinese advanced manufacturing industry. The main concepts corresponding to this research question are the influencing factors of SCC, and the relationship between the influencing factors of SCC. Our second question is what state of the main influencing factors will lead to the highest level of SCC in China's advanced manufacturing industry. The main concept corresponding to this research question is the effect of the influencing factors of SCC. The research content of this concept is included in the research related to the influence factors of SCC. Therefore, we started with a literature review of the influencing factors of SCC and the relationship between the influencing factors of SCC. In addition, we pondered whether the literature review of these two parts completely draws out the gaps in the research questions. Our feel no. The two parts of the literature review mentioned reveal that the research gaps corresponding to the research questions are: on the one hand, there are research gaps in the system of influencing factors of SCC in a comprehensive and widely used scope. On the other hand, there is a lack of comparative aspects of the magnitude of influence relationships among the influencing factors of SCC. They did not reveal that the influence factors of SCC do not conform to the characteristics of synergistic influence. Therefore, we added the literature review corresponding to this part of the gap, i.e., the research on the application of the theory of comparative advantage and synergetics. By adding the content and application of synergetics, we reveal that the findings of previous studies on the influencing factors do not apply the synergistic influence characteristics of synergetics; by analyzing the connection between the theory of comparative advantage and synergetics, we reveal the gaps in the combination of the two in previous studies. Finally, we have formed the analysis idea of “influencing factors of SCC → relationship between influencing factors of SCC → application of the theory of comparative advantage and synergetics”. 

In addition, according to your guidance, in order to ensure that the literature review is fully focused on our research topic, we focus on the latest research on the main variables in each paragraph of the literature review on “influencing factors of SCC”. 

In terms of details, the literature reviewed is from the last three years in order to adequately represent the latest research. 

Importantly, in the final section of the literature review, we identified research gaps by analyzing the findings of previous researchers. Subsequently, we relate them to the research objectives. This further highlights the relevance of the latest research status to the research problem-solving process and its results. In addition, during the literature combing process in each section, we have briefly reviewed the literature in that section. To highlight the revisions, we have marked the revisions in the manuscript in red font (L143-338 in red). 

Comment 3: Consluson section is too long. Divide it into “discussion” and “conclusions” sections. In the Discussion section, the obtained results should be confronted with the opinions contained in the literature on the subject and with the results of the other studies, so that it is clear whether the obtained results confirm or question the current state of knowledge. In the “conclusion” section, on the other hand, focus on the final conclusion, research limitations, implications and future research directions. 

Response 3: We are very grateful for your suggestions. Firstly, we have added a discussion section based on your suggestions. We have included in the discussion section two new insights distilled from the content of the empirical studies. We have compared the new insights with their relevant studies separately. The results of this comparison confirm our new insights. In addition, we conclude the discussion section with a discussion of the circular alternatives to which these new insights can be applied, further demonstrating the benefits of this study and these new insights. Secondly, we have reworked the conclusion section. We have highlighted the conclusions of the empirical study by summarizing our work, which is a combination of background theory, data theory/analysis and key findings in response to the questions posed in the introduction. Some inferences to the key findings are also analysed following the conclusion. The implications of the conclusion section we have divided into two parts: theoretical implications and management implications. For the theoretical implications, we reason about the two values of the thesis and highlight novelty by comparing them with previous studies and industry data respectively. Based on the empirical evidence showing competitive advantage as a major influencing factor, we highlight the development of management insights in the management insights section by presenting recommendations around managers' management activities and comparing them with previous measures. In terms of research limitations, we present recommendations for future research by analyzing the limitations of our study and making informed recommendations from comparisons with previous research. (marked in red L795-992).

Reviewer #2:

We appreciate your interest and insightful comments. These comments are critical to the improvement of the manuscript. We strongly agree with your suggestions, and we are keenly aware of the importance of improvement in terms of grammar, spelling, and presentations. As a result, we have made careful English polishing to the manuscript. Also, we have fixed minor errors in the methods section of the previous manuscript and marked them in red. We hope that our corrections will be accepted. If there are still areas that need to be corrected, we look forward to hearing from you and we will certainly endeavor to do our best to improve them.

Comment 1: The title should be changed to reflect more clearly the contribution of the study. 

Response 1: Thank you for your guidance and very important suggestions. Based on your guidance, we have first carefully rethought and confirmed our research contributions. One of our theoretical contributions is the establishment of a triadic motivational framework of “competitive advantages - interest demands - cooperative relationships” based on the theory of comparative advantage and synergetics for the synergy of socio-economic systems. This framework is the first to establish the link between the theory of comparative advantage and synergetics. Its components satisfy the logical condition of “behind-the-scenes” domination of the system by sequential parameters in synergetics, which is not achieved by the literature on synergistic influences. To reflect the research contributions arising from this triadic motivational framework, we have briefly reflected its elements, namely competitive advantages, benefit demands, and cooperative relationships, in the title. In addition, we have enriched the previous one-way analysis between interests and advantages and the influence of interests and advantages on SCC through comparative analysis, highlighting the overwhelming role of competitive advantages in upgrading the level of SCC in advanced manufacturing industries. This points out the primary and secondary tone of SCC in advanced manufacturing for the first time and explains economic phenomena that have not been analyzed in the literature. To reflect this research, we emphasize the primary and secondary influences in the title. Therefore, we refreshed the title to Competitive Advantage, Relationship and Benefit: Primary and Secondary Influencing Factors of Supply Chain Collaboration in China's Advanced Manufacturing Industry. we also revised and refreshed the significance of the study in the conclusion section accordingly in this reflection process. To highlight the revisions, we have marked the revisions in the manuscript in red font (red labeled L1-3). 

Comment 2: Please revise the Conclusion in the abstract to avoid overly casual language.

Response 2: Thank you sincerely for your suggestion. In response to your suggestions, we have reworked the content of the conclusions in the abstract. Equally important, we highlight the significance of the study in the abstract and supplement the first half of the abstract with a statement of the research gaps and real-world issues corresponding to the significance of the study. To highlight the changes, we have marked them in red in the manuscript (L16-43 in red). 

Comment 3: In the Introduction, you need to connect the state of the art to your paper goals.

Response 3: Thank you very much for your constructive guidance. Based on your suggestions, we have reworked the way the questions are presented in the introduction. The content is as follows. 

In fact, it is still difficult to bring about the full implementation of high-level SCC. On one hand, factors affecting SCC, such as the level of information technology [5], environmental protection tendency [6], and profit distribution mechanism [7] are spread throughout the supply chain system. This makes it difficult to focus on the main direction of SCC. On the other hand, synergetics do not indicate the specific factors affecting the synergy but only point out that the sequential parameters were significant and root signs before and after the system phase change and guided the fast variables to influence the synergy of the system [8]. So, in the absence of theories indicating the key specific factors for achieving SCC, many advanced manufacturing companies are at their wits’ end when facing the unprecedentedly complex situation. To effectively cope with the problems caused by their own shortcomings, trade frictions, and the COVID-19 pandemic, Chinese advanced manufacturing supply chains need to find and seize the key to realizing SCC. Therefore, the purpose of this study corresponds to the following questions:

1. What are the primary and secondary influencing factors of SCC in China’s advanced manufacturing industry, and what are their relationship mechanisms?

2. In terms of their effect, what state of the main influencing factors will lead to the highest level of SCC in China’s advanced manufacturing industry?

Comment 4: In the Introduction, the 2nd paragraph must support a practical example of the problem descriptions. A potential suggestion to the author/s is to support the importance of a title with some practical examples. However, too many things are bundled together, so the logic and results are not very clear or convincing.

Response 4: Thank you very much for your careful guidance. As for the case subject, based on your suggestion, we chose the case of Hisense, a Chinese advanced manufacturing company, and its partner's SCC. The reason we chose this case is that it involves a leading company in its industry and the case happened very recently, which is representative of the current SCC in advanced manufacturing industry. In terms of case content, in order to ensure that the case supports the problem description, we not only ensured that the case belongs to the field of advanced manufacturing SCC, but also pointed out the motivation of SCC in the first part of the case. In terms of case summary and analysis, we link the cases to the research topic. Equally important, in order to avoid the lack of clarity in logic caused by putting too many things together, we use the logic in the second paragraph: “SCC is deeply affected by the situation inside and outside the supply chain.” So the case is introduced, and then the case is linked to the research topic, which leads to the reality of the problem, i.e., “It is still difficult to achieve a high level of supply chain synergy on the ground.” By reviewing the literature on the real-world problem, the theoretical gaps in the problem are identified and followed by the research question. The content is as follows. 

SCC is deeply affected by the situation inside and outside the supply chain. Logistic disruptions and lack of capacity supply during the pandemic caused a series of operational problems. To optimize business processes, improve rapid response capabilities, increase profits reduced by stagnant sales, achieve close collaborative relationships, improve trust levels, and so forth, in 2021, Hisense and JD upgraded the traditional “production by order” to “production by plan” based on market demand through deep SCC such as planning collaboration, procurement, distribution and return data collaboration, and inventory layout optimization. This has greatly improved supply chain efficiency and met their goal of laying out SCC. As we can see, their numerous purposes of laying out SCC influence their layout measures, which, in turn, influence the level of SCC they are able to achieve. Their ability to achieve a higher level of SCC is due to the fact that their layout measures are related to the main influencing factors of SCC. In fact, it is still difficult to bring about the full implementation of high-level SCC. On one hand, factors affecting SCC, such as the level of information technology [5], environmental protection tendency [6], and profit distribution mechanism [7] are spread throughout the supply chain system. This makes it difficult to focus on the main direction of SCC. On the other hand, synergetics do not indicate the specific factors affecting the synergy but only point out that the sequential parameters were significant and root signs before and after the system phase change and guided the fast variables to influence the synergy of the system [8]. So, in the absence of theories indicating the key specific factors for achieving SCC, many advanced manufacturing companies are at their wits’ end when facing the unprecedentedly complex situation. To effectively cope with the problems caused by their own shortcomings, trade frictions, and the COVID-19 pandemic, Chinese advanced manufacturing supply chains need to find and seize the key to realizing SCC. Therefore, the purpose of this study corresponds to the following questions:

1. What are the primary and secondary influencing factors of SCC in China’s advanced manufacturing industry, and what are their relationship mechanisms?

2. In terms of their effect, what state of the main influencing factors will lead to the highest level of SCC in China’s advanced manufacturing industry?

Comment 5: Please follow the literature review by a clear and concise state of the art analysis. This should clearly show the knowledge gaps identified and link them to your paper goals. Please reason both the novelty and the relevance of your paper goals. Clearly discuss what the previous studies that you are referring to.What are the Research Gaps/Contributions?

Response 5: We greatly appreciate these points of your guidance. Based on these comments, we have done the following. First, in the last part of the literature review, we clearly and briefly indicate the usefulness of the previous studies for us. Second, we focused on three aspects of the previous research and used them as a relevant basis for our study. The literature analysis of each viewpoint is followed by a comparative presentation of the missing research content relevant to our research goal, and is immediately followed by a statement of “therefore” revealing the knowledge gap about the study goal. This achieves a combination of research gaps and study goals. Finally, we reveal in two sentences the space reserved by the lack of research for the novelty of our research objectives. It is also important to note that we end each section of the literature review with a brief review of the literature from that section (i.e., at the end of each paragraph/section). This keeps the literature review process in line with the goals of the paper. Finally, it is important to note that we also briefly repeat the work done in the last part of the literature review (2.4 Research gap) in the introduction section. The process is as follows: we follow the examples in the second paragraph with a brief review of the relevant literature on real-world issues related to the research objectives. We then point out the gaps related to the thesis objectives and formulate the research questions accordingly. After the research question, we state our research and indicate the novelty of the research content. In order to highlight the revisions, we have marked them in red in the manuscript (L311-338 in red). 

Comment 6: There is no flow in the text. It partly depends on the lack of proofreading but also on the fact that many statements and claims are made without being followed up by a clear and logical discussion. I suggest the authors to add more literature to make the literature strong. In this study, literature part, the conceptualization of each study variable can be added. Additionally, explain the relationship between the variable of the study. Please explains the use of theory to develop this model. Literature Review has the chance to be further improved: it seems that the authors have made the retrospection. However, via the review, what issues should be addressed? What is the current specific knowledge gap? What implication can be referred to? The above questions should be answered. Authors need to propose their study.

Response 6: We appreciate your suggestions for writing in the literature review section. In order to ensure coherence and logic and to explain the relationship between the research variables, we have organized the structure of the literature review section. Our structure is mainly based on our research questions. First, our first research question is what are the main and secondary influencing factors and their relationship mechanisms of SCC in China's advanced manufacturing industry, which corresponds to the main concepts of influencing factors of SCC and relationship of influencing factors of SCC. Our second question is what state of the main influencing factors will lead to the highest level of SCC in China's advanced manufacturing industry, and its corresponding main concept is the influence effect of the influencing factors of SCC. The study of the effect of influence factors is included in the research related to the influence factors of SCC. Therefore, we first sort out the literature on the influencing factors of SCC and the relationship between the influencing factors of SCC. In addition, we found that these two parts of the literature review do not lead to the gaps corresponding to the full research questions. They only reveal the research gaps in the system of influencing factors of SCC and comparison of the magnitude of the relationship between influencing factors of SCC corresponding to the research questions. They did not reveal the problem of previous studies on the influencing factors of SCC that do not match the characteristics of the role of synergistic influence. Based on this, we added a literature review related to the research on the application of the theory of comparative advantage and synergetics. This part not only reveals that previous studies on the influence factors do not apply the synergistic influence characteristics of synergetics, but also reveals the gaps in the combination of the theory of comparative advantage and synergetics and the basis of their development through the analysis of the linkage between them. Finally, we have formed the analysis idea of “Influencing factors of SCC → Relationship between factors influencing SCC in manufacturing → Application of the theory of comparative advantage and synergetics”.

In terms of conceptualization of each research variable: According to your guidance, in order to ensure that the literature review is fully focused on our research topic, we have focused on the latest research on the main variables in each paragraph of “2.1 Influencing factors of SCC” and have given conceptual explanations for each variable.

In terms of details: in order to adequately represent the latest research, we have used literature from the last three years in writing the literature review. In order to enrich the literature, we followed your suggestion of adding more literature.

For the questions addressed in the literature review, the research gaps and their implications: in the last part of the literature review, we sorted out the research gaps based on the findings of previous researchers and linked them to the research objectives after the research gaps. This further highlights the relevance of the state of the art to the research problem-solving process and its results. We first focus on three areas to discuss previous research and use them as a relevant basis for our study, which demonstrates the problems that the literature review has addressed for our unfolding research. Second, after this, we present the missing research content relevant to study objectives and reveal the knowledge gaps regarding our study objectives, in the process we combine the research gaps and the study objectives. Finally, we reveal in two sentences the space of the novelty for us to achieve our goals, which indicates what the current research gaps imply. In fact, we conclude each section of the literature review with a brief review of the content of that section. This is to keep the literature review process close to the paper's objectives.

Regarding the theoretical interpretation of the development of this model: we consider the literature review as a way to sort out the research results related to the research question and to point out the research gaps. The ternary motivation model we developed is not a previous research result and it can only be considered as a research gap and implies a research contribution. Therefore, we do not believe that the literature review section is appropriate to reveal how the two theories were used to develop this model. The literature review section is only suitable to elaborate on the relevant presentations of the two underlying theories involved in developing the model, the applied research, and the combined basis/commonalities between the two theories, which can provide a basis for explaining the use of the two theories in developing the model. In contrast, we believe that the explanation of how to use the theory development model should be placed in the theoretical analysis section. The theoretical analysis explains how to use the theory to develop the model and apply the model developed based on the two theories and the content of synergetics to the analysis of the advanced manufacturing supply chain to achieve the effect of validation through analysis. For this reason, we remove the theoretical content originally located above “Figure 2” and place it at the beginning of the theoretical analysis section. In addition, we consider that the research sequence generally consists of a theoretical analysis part in which the hypothesis is formulated and followed by an empirical approach to test the hypothesis. Therefore, we present the hypothesis in the end of the theoretical analysis. This takes up exactly the theoretical analysis of the influencing factors of supply chain synergy in advanced manufacturing industry above, and also responds to the model hypothesis in the empirical phase. Accordingly, we supplement the hypothesized results in the empirical analysis section.

In order to highlight the revisions, we have marked the revisions in red in the manuscript (marked in red as L143-371, L508-514, L695, L714-715).

Comment 7: Page11 manuscript, line 229 to 227, authors have made a good linkaged among the literature. Here I would suggest, where author discuss about the innovation capacity, here, I recommend that add the following with innovation. I encourage authors to carefully consider and incorporate the suggested articles for the manuscript quality improvements and justify when necessary.

Response 7: We are very grateful for your endorsement and advice on this section. Thank you very much for the instructive articles you have shared. Through reading these two articles, we have discovered the relationship between the level of SCC and innovation performance, and also found the role and influence of social performance between them. These are realizations that we had not previously understood or considered. So these two articles are very informative for our analysis of innovation in advanced manufacturing. We have analyzed the importance of competitive advantages for advanced manufacturing supply chains and the shortcomings of advanced manufacturing firms in terms of innovation. Therefore, the relationships between SCC, social performance and innovation performance revealed in these two articles become the breakthrough point for us to analyze the improvement of innovation level in advanced manufacturing companies. Therefore, we use “SCC-innovation performance” and “SCC-social performance-innovation performance” as the two main lines for analyzing the improvement of innovation level respectively. In addition, these two lines of thought are analyzed to support and highlight the role of competitive advantages represented by innovation in SCC. This validates the influence of competitive advantages that we have mentioned. This gives our analysis a back-and-forth and self-validating effect. To highlight the revisions, we have marked the revisions in the manuscript in red font (L410-417 in red). 

Comment 8: Page 25, 467- to 475.I like the way author organize the paper, here author discuss about the advanced manufacturing. It is up to the authors, but it will be good, if they consider, and linked advance manufacturing concept with the industry 4.0 and additive manufacturing. Here, I suggest the following article.This article has highlighted the advance manufacturing.

Response 8: We appreciate your endorsement of this section and your suggestions for expanding our thinking. Thank you very much for these two articles. They have deepened our understanding of Industry 4.0 (I4.0), circular economy, and additive manufacturing. After reading Transition toward green economy: Technological Innovation's role in the fashion industry, we found that new digital technologies contribute to a green and smart transition in the manufacturing sector. For example, the development of new technologies for sustainability brought about by technological innovation can have a noticeable impact on reducing waste in the fashion industry, which generates a lot of waste. After studying Designing Value Chains for Industry 4.0 and a Circular Economy: A Review of the Literature, we also found that the combination of additive manufacturing technology and digital technology can help advanced manufacturing obtain a more advanced production model, resulting in the production of Industry 4.0-standard products. These insights allow us to establish a connection between advanced manufacturing, I4.0, and Industrial Internet-related technologies, which helps us to propose a clearer example for the construction of competitive advantages in the advanced manufacturing supply chain. Therefore, we propose in the paper the impact of new technologies on the development model of advanced manufacturing in an environment where the Industrial Internet is vigorously developed. One example is used as a proof. (To highlight the revisions, we have marked the revisions in the manuscript in red font (red markings are L701-707)). 

Comment 9: The methods and results show good explanation.As authors know that, method sections determine the results. I suggest refining these sections to remove minor errors. Add justification of this study sample. Kindly focus on the following suggestions

1-how the study was design, discuss complete data collection process, and compare with other methodologies.

2-how the study was carried out, here, highlight the detailed sampling procedures,and pilot testing procedures, if any adopted.

3-How data were analyzed.

Response 9: We are very sorry for our error. We have checked this section in detail for minor errors and fixed them. The following corrections have been made in response to the comments.

There are no pre-tests in our study. The reason is as follows: a pre-test is a pre-experiment carried out on the feasibility of a research design. The Haken model, on which our study was developed, is a model with its roots in synergetics and is theoretically feasible. In addition, the practical feasibility of the model is also mentioned in our introduction to the Haken model (the model is widely used in research in various subject areas, such as the evolutionary analysis of industrial systems and supply chain systems.) Our article therefore does not require pre-tests to determine the feasibility of the study.

The process of research design is described below. To ensure the soundness of the study design, we explain the reasons for the choice of the model and supplement the introduction of the model in the 4.1 Haken model section. Then we started with the main elements of the study and the methods used in the study, identifying the shortcomings and inapplicability of methodologies such as the questionnaire method and the structured interview method in 4.2 Sample selection, index system, and data processing. This achieves a comparison with other methodologies. In terms of the rationale for the research sample, after comparison with other methodologies and identification of the empirical analysis approach, we determined the sampling approach based on the data source and based on the research subjects. We determined the sampling method as judgmental sampling. Subsequently, we determined the sample for the study through the determination of the industry scope, the theoretical requirements for establishment years of firms, and the determination of the indicators for judging representative companies. The complete data collection process is then discussed. In terms of data analysis, we detail the data pre-processing and entry process into the software, and the operation of data analysis in the software is also described in detail.

To highlight the revisions, we have marked the revisions in the manuscript in red font (red labeled L518-526, L531-536, L577-590, L602-616).

Comment 10: The discussion needs to be more elaborate and the author should also compare it with previous research in this area. What are the main benefits of this study? The same goes for recommendations and future research.

Response 10: We would like to thank you for your suggestions for the discussion section. Based on your suggestions, we have written the discussion section. We have separately compared the new insights gained from the empirical study with its related research, which confirms our new insights. To demonstrate the benefits of this research and these new insights, we conclude the discussion section by discussing circular alternatives to the application of these new insights. In addition, we analyze the management enlightenment and research limitations in the conclusion with previous studies to achieve the development of management enlightenment and the purpose of proposing future research. (L769-860, L892-992 marked in red).

Comment 11: The conclusion is very weak.Please make sure your conclusions section underscores the scientific value-added of your paper, and/or the applicability of your findings/results. Highlight the novelty of your study. In addition to summarizing the actions taken and results, please strengthen the explanation of their significance. It is recommended to use quantitative reasoning comparing with appropriate benchmarks, especially those stemming from previous work.It should also be an extrapolation of the key findings from the research and not a summary. So, there should be conclusions around the background theory, data theory/analysis and, key outcomes. The authors should have included the following sub-sections within the conclusion section with more details:

-The authors should offer implications for theory and practice separately as discussed above. See suggested articles to revise the implications and offer actionable points for implementing the offered implications.

--I suggest the authors refine the managerial insights based on the findings

- Limitations in the suggested approach should be discussed in the conclusions section. Implications for future research may also be included in the Conclusion at the end.

Response 11: We are very grateful for your suggestions and have carefully revised the conclusions. First, we summarize the conclusions based on the work done, background theory, data theory/analysis and key findings. And the inferences proposed for the conclusions. Secondly, in order to ensure the scientific value added to the paper and the applicability of the findings/results, we have carefully considered and collated the theoretical contributions of the study, one of which is our development of a triadic motivational framework of “competitive advantages — Interest demands — cooperative relationships” for synergy of socio-economic system. The other is that we provide the first direction for advanced manufacturing practitioners to improve supply chain synergies and explain economic phenomena that have not been analyzed in the literature. In doing so, we use both comparative literature and quantitative comparative reasoning. Our refines the novelty through these two analyses. Third, based on previous studies, we open new paths and elaborate specific measures for managers to carry out management activities with the help of the identified main influencing factors of SCC in the management inspiration. Finally, we point out the direction of future research and its implications based on the content of previous studies and the shortcomings of our study. (L862-992 marked in red).

Comment 12: The writing of the paper needs a lot of improvement in terms of grammar, spelling, and presentations. The paper needs careful English polishing since many typos and poorly written sentences exist. * Check the usage of the commas carefully. I'm not going to keep identifying problems with the writing, but please do not just fix these examples and leave the rest - find them all and fix them all. I suggest you have someone fluent in English go over the whole paper and check each sentence.Avoid repetitions. I can see several repetitions at different places in this paper. Thorough proofreading is required.

Response 12: Thank you very much for your advice on writing. Based on your advice, we had a fluent English teacher go through the final version of our essay and have him focus on typos, comma usage, and repetitive sentences throughout the essay. These efforts ultimately optimized our essay for grammar, spelling, and presentation issues. (We have highlighted in red in the revised version, and the markings are spread throughout the text).

---

## [Decision Letter · Decision Letter 1]

26 Dec 2022

PONE-D-22-25768R1Competitive Advantage, Relationship, and Benefit: Primary and Secondary Influencing Factors of Supply Chain Collaboration in China’s Advanced Manufacturing IndustryPLOS ONE

Dear Dr. Lin,

Thank you for submitting your manuscript to PLOS ONE. After careful consideration, we feel that it has merit but does not fully meet PLOS ONE’s publication criteria as it currently stands. Therefore, we invite you to submit a revised version of the manuscript that addresses the points raised during the review process.

ACADEMIC EDITOR:  

We look forward to receiving your revised manuscript.

Kind regards,

Muhammad Ikram

Academic Editor

PLOS ONE

Journal Requirements:

Reviewers' comments:

Reviewer's Responses to Questions

**Comments to the Author**

1. If the authors have adequately addressed your comments raised in a previous round of review and you feel that this manuscript is now acceptable for publication, you may indicate that here to bypass the “Comments to the Author” section, enter your conflict of interest statement in the “Confidential to Editor” section, and submit your "Accept" recommendation.

Reviewer #1: All comments have been addressed

Reviewer #2: All comments have been addressed

2. Is the manuscript technically sound, and do the data support the conclusions?

Reviewer #1: Yes

Reviewer #2: Yes

3. Has the statistical analysis been performed appropriately and rigorously? 

Reviewer #1: Yes

Reviewer #2: Yes

4. Have the authors made all data underlying the findings in their manuscript fully available?

Reviewer #1: Yes

Reviewer #2: Yes

5. Is the manuscript presented in an intelligible fashion and written in standard English?

Reviewer #1: Yes

Reviewer #2: Yes

6. Review Comments to the Author

Reviewer #1: The manuscript in its present form has been revised augmenting its quality. All my concerns are properly addressed and I have no more suggestions.

Reviewer #2: - I appreciate the novelty of the author's contributions, but still, authors need to grasp good ideas on developing the research gap in the Introduction. The manuscript concerning the presentation of ideas shows improvement, but there are still some sentence structure changes and develop more clear research gap with the support of previous published studies in the domain of your study aim. This is a beautifully analyzed and written paper with an important message. I would recommend publication after the issues below are addressed.Paper is in its well form, for further, I suggest the following. Impact of Social Supply Chain Practices on Social Sustainability Performance in Manufacturing Firms, International Journal of Innovation and Sustainable Development, Vol. 13, No. 2, 2019. DOI: 10.1504/IJISD.2019.098996

-The last paragraph of the Introduction must be done a summary (resume) of the paper, i.e., a clear idea about what will be studied in the paper. I could not identify this. The last paragraph's purpose in the Introduction section is to summarize the main points, restate the paper's main idea, and show how the paper statements were proven. I think the paper will considerably improve and be a highly cited article.

-Check the citations and references (one by one) if there is any missing information. Citations and references must be 100% accurate according to the journal guidelines.

-Moreover, please check all the references in the text carefully.

7. PLOS authors have the option to publish the peer review history of their article (what does this mean?). If published, this will include your full peer review and any attached files.

Reviewer #1: No

Reviewer #2: No

---

## [Author Response · Author response to Decision Letter 1]

9 Feb 2023

Response to Editor

Comment 1: Please review your reference list to ensure that it is complete and correct. If you have cited papers that have been retracted, please include the rationale for doing so in the manuscript text, or remove these references and replace them with relevant current references. Any changes to the reference list should be mentioned in the rebuttal letter that accompanies your revised manuscript. If you need to cite a retracted article, indicate the article’s retracted status in the References list and also include a citation and full reference for the retraction notice. 

Response 1: We are very grateful for these suggestions. They enhance the rigor and persuasiveness of the article. So we checked the reference list very carefully. We found that we did not cite the retracted article. But we found the problem of double citations. Therefore, we deleted the repeated references. Finally, we reordered the references according to the order of the citations in the text. To highlight the revisions, we have marked the revisions in the manuscript in red font (L1016-1253 in red). 

 

Response to Reviewers

Reviewer #1:

Thank you very much for your positive and complimentary comments on the content of our article. It is good to know that all your concerns are properly addressed and you have no more suggestions. This gives us very great encouragement and support. We will continue to work hard. Thank you again for your recognition！

Reviewer #2:

Thank you for your insightful comments and careful guidance. Now we have carefully analyzed and revised exactly according to your comments. With the help of previous published studies, we revised the sentences and redeveloped the research gap in Introduction section. After deleting duplicate references, we have adjusted the reference list according to the added citations. In addition, we add the main points, restate the paper's main idea, and show how statements of the paper were proven in the last paragraph of Introduction section. After receiving your valuable comments, the level of our article gets great improvement.

Comment 1: authors need to grasp good ideas on developing the research gap in the Introduction. The manuscript concerning the presentation of ideas shows improvement, but there are still some sentence structure changes and develop more clear research gap with the support of previous published studies in the domain of your study aim. This is a beautifully analyzed and written paper with an important message. I would recommend publication after the issues below are addressed.Paper is in its well form, for further, I suggest the following. Impact of Social Supply Chain Practices on Social Sustainability Performance in Manufacturing Firms, International Journal of Innovation and Sustainable Development, Vol. 13, No. 2, 2019. DOI: 10.1504/IJISD.2019.098996. 

Response 1: Thank you sincerely for your suggestion. In response to your suggestions, we have carefully read the research gap in the Introduction section of your proposed paper. We found that the paper you presented provided very much previous published studies in the development of the research gap. This is in line with your request to support and develop a clearer research gap with past research. In addition, we found that the research gap can be followed by a research question and an elaboration of research implications. Thank you very much for suggesting this article. By reading Impact of Social Supply Chain Practices on Social Sustainability Performance in Manufacturing Firms, it became clear how the research gap was formed and the logic between the research gap and the rest of the introduction. Therefore, we analyzed the current state of the research based on the three theoretical innovations of the paper before the research question. This process focused on adding past studies. We found an idea in the article you presented that could provide support for our analysis, so we also cited the article you presented. Finally, we developed a clearer research gap. We have also embellished the language in this section. To highlight the revisions, we have marked the revisions in the manuscript in red font (red labeled L77-105). 

Comment 2: The last paragraph of the Introduction must be done a summary (resume) of the paper, i.e., a clear idea about what will be studied in the paper. I could not identify this. The last paragraph's purpose in the Introduction section is to summarize the main points, restate the paper's main idea, and show how the paper statements were proven. I think the paper will considerably improve and be a highly cited article.

Response 2: Thank you very much for your constructive guidance. We have revised the section in order according to your suggestions. First, we added the main content of the article at the beginning of the last paragraph of the introduction. That is, what will be studied in this article. Secondly, in this paragraph, our previous introduction to each part of the article clearly expresses how the ideas of the article are proven. Therefore, we only add the formulation of hypotheses to make the content clearer. Finally, we add the main ideas/results of the article at the end of this paragraph. To highlight the changes, we have marked them in red in the manuscript (L143-162 in red). 

Comment 3: -Check the citations and references (one by one) if there is any missing information. Citations and references must be 100% accurate according to the journal guidelines.-Moreover, please check all the references in the text carefully.

Response 3: Thank you very much for your careful guidance. Based on your suggestions, we have carefully checked citations and references to avoid missing information and to ensure that citations and references correspond. In addition, the format of citations and references in this article is strictly in accordance with the journal guidelines. Items 71 and 72 of the references cite web links. Since we see that the format of web links is not defined in the journal guidelines, and we have seen web links placed in the reference list in other articles. So we put the web links directly. In order to highlight the revisions, we have marked them in red in the manuscript (L1016-1253 in red).

---

## [Decision Letter · Decision Letter 2]

30 Mar 2023

PONE-D-22-25768R2Competitive Advantage, Relationship, and Benefit: Primary and Secondary Influencing Factors of Supply Chain Collaboration in China’s Advanced Manufacturing IndustryPLOS ONE

Dear Dr. Lin,

Thank you for submitting your manuscript to PLOS ONE. After careful consideration, we feel that it has merit but does not fully meet PLOS ONE’s publication criteria as it currently stands. Therefore, we invite you to submit a revised version of the manuscript that addresses the points raised during the review process.

We look forward to receiving your revised manuscript.

Kind regards,

Shi Yin

Academic Editor

PLOS ONE

Journal Requirements:

Additional Editor Comments:

The manuscript has significantly improved as compared to the previous version. Indeed, the authors tried to improve it, and the main weaknesses are solved.

Please revise your paper according to comments and resubmit the manuscript.

1.The first letter of the word in Figure1 should be capitalized.

2.Please enhance the clarity of Figure 3.

Reviewers' comments:

Reviewer's Responses to Questions

**Comments to the Author**

1. If the authors have adequately addressed your comments raised in a previous round of review and you feel that this manuscript is now acceptable for publication, you may indicate that here to bypass the “Comments to the Author” section, enter your conflict of interest statement in the “Confidential to Editor” section, and submit your "Accept" recommendation.

Reviewer #1: All comments have been addressed

Reviewer #2: All comments have been addressed

2. Is the manuscript technically sound, and do the data support the conclusions?

Reviewer #1: (No Response)

Reviewer #2: Yes

3. Has the statistical analysis been performed appropriately and rigorously? 

Reviewer #1: Yes

Reviewer #2: Yes

4. Have the authors made all data underlying the findings in their manuscript fully available?

Reviewer #1: Yes

Reviewer #2: No

5. Is the manuscript presented in an intelligible fashion and written in standard English?

Reviewer #1: Yes

Reviewer #2: Yes

6. Review Comments to the Author

Reviewer #1: I would like to thank the authors for the changes and comments introduced in the article.

I have no more comments.

Reviewer #2: The authors have done extensive revision and addressed all the comments to my satisfaction. I am happy to recommend accepting the revised version of this manuscript.

7. PLOS authors have the option to publish the peer review history of their article (what does this mean?). If published, this will include your full peer review and any attached files.

Reviewer #1: No

Reviewer #2: No

---

## [Author Response · Author response to Decision Letter 2]

9 Apr 2023

Response to Editor

Thank you very much for your approval of the content of our article. Now we have carefully revised the two figures you mentioned and the reference list based on your comments. We hope our modifications will be approved by you.

Comment 1: The first letter of the word in Figure1 should be capitalized. 

Response 1: We are very grateful for the suggestion. We capitalize the first letter of each word in Figure 1 and place the redrawn Figure 1 in the manuscript. We believe that the manuscript is more in line with the journal's specifications with this modification(red labeled L424). 

Comment 2: Please enhance the clarity of Figure 3. 

Response 2: Thank you very much for your careful guidance. We redrew Figure 3 according to the image format required by the journal and placed it in the manuscript. The clarity of Figure 3 is effectively improved. In addition, files of Figures 1, 2, and 3, which conform to the journal's format requirements, were uploaded in the manuscript submission system separately again(red labeled L812). 

Comment 3: Please review your reference list to ensure that it is complete and correct. If you have cited papers that have been retracted, please include the rationale for doing so in the manuscript text, or remove these references and replace them with relevant current references. 

Response 3: Thank you sincerely for your suggestion. Once again, we carefully checked the reference list. We found errors in the writing of some names, so we immediately corrected these errors. We are sorry for these errors. In addition, we have not cited papers that have been retracted. To highlight the revisions, we have marked the revisions in the manuscript in red font (red labeled L1052, L1055, L1085, L1108, L1148, L1184, L1187, L1191).

---

## [Editor Report · Decision Letter 3]

19 Apr 2023

Competitive Advantage, Relationship, and Benefit: Primary and Secondary Influencing Factors of Supply Chain Collaboration in China’s Advanced Manufacturing Industry

PONE-D-22-25768R3

Dear Dr. Lin,

We’re pleased to inform you that your manuscript has been judged scientifically suitable for publication and will be formally accepted for publication once it meets all outstanding technical requirements.

Kind regards,

Shi Yin

Academic Editor

PLOS ONE
---

## [Editor Report · Acceptance letter]

29 May 2023

PONE-D-22-25768R3 

Competitive Advantage, Relationship, and Benefit: Primary and Secondary Influencing Factors of Supply Chain Collaboration in China’s Advanced Manufacturing Industry 

Dear Dr. Lin:

I'm pleased to inform you that your manuscript has been deemed suitable for publication in PLOS ONE. Congratulations! Your manuscript is now with our production department. 

Kind regards, 

on behalf of

Professor Shi Yin 

Academic Editor

PLOS ONE